# A single-cell atlas of the cycling murine ovary

**Mary E Morris[1]\*[†], Marie-Charlotte Meinsohn[2,3][†], Maeva Chauvin[2,3], Hatice D Saatcioglu[2,3], Aki Kashiwagi[2,3], Natalie A Sicher[2,3], Ngoc Nguyen[2,3], Selena Yuan[2,3], Rhian Stavely[2,3], Minsuk Hyun[4], Patricia K Donahoe[2,3], Bernardo L Sabatini[4], David Pépin[2,3]\***

[1]Department of Gynecology and Reproductive Biology, Massachusetts General Hospital, Boston, United States; [2]Pediatric Surgical Research Laboratories, Massachusetts General Hospital, Boston, United States; [3]Department of Surgery, Harvard Medical School, Boston, United States; [4]Howard Hughes Medical Institute, Department of Neurobiology, Harvard Medical School, Boston, United States

**Abstract** The estrous cycle is regulated by rhythmic endocrine interactions of the nervous and reproductive systems, which coordinate the hormonal and ovulatory functions of the ovary. Folliculogenesis and follicle progression require the orchestrated response of a variety of cell types to allow the maturation of the follicle and its sequela, ovulation, corpus luteum formation, and ovulatory wound repair. Little is known about the cell state dynamics of the ovary during the estrous cycle and the paracrine factors that help coordinate this process. Herein, we used single-cell RNA sequencing to evaluate the transcriptome of >34,000 cells of the adult mouse ovary and describe the transcriptional changes that occur across the normal estrous cycle and other reproductive states to build a comprehensive dynamic atlas of murine ovarian cell types and states.

## Editor's evaluation

This manuscript presents an important and useful dataset for understanding cellular and transcriptional dynamics during the estrous cycle in mice. Using single-cell RNA sequencing, the authors' data is compelling, providing new marker genes for different cell types. These data will be useful for understanding ovarian biology and will be of interest to biologists studying other tissues.

**\*For correspondence:**
MEMORRIS@mgh.harvard.edu (MEM);
dpepin@mgh.harvard.edu (DP)

[†]These authors contributed equally to this work

**Competing interest:** The authors declare that no competing interests exist.

## Introduction

The ovary is composed of a variety of cell types that govern its dynamic functions as both an endocrine organ capable of producing hormones such as sex steroids and a reproductive organ orchestrating the development of follicles, a structure defined by an oocyte surrounded by supporting somatic cells such as granulosa cells and theca cells. Most follicles in the ovary are quiescent primordial follicles, representing the ovarian reserve. Once activated, a primordial follicle grows in size and complexity as it progresses to primary, preantral, and antral stages, adding layers of granulosa and theca cells and forming an antral cavity, until it ultimately ejects the oocyte-cumulus complex at ovulation while the follicular remnants undergo terminal differentiation to form the corpus luteum (CL) (*Dunlop and Anderson, 2014*). This process necessitates precise coordination of germ cells and several somatic cell types, including granulosa cells, thecal cells, vascular cells, and other stromal cells of the ovary to support the growth of the oocyte until its ovulation or, as is most often the case, undergo follicular atresia. In addition to supporting germ cells, ovarian somatic cells must produce the necessary hormonal cues, as well as coordinate the profound tissue remodeling, necessary to accommodate

these dynamic developing structures. For reproductive success to occur, the state of each of these cells must change in a coordinated fashion over the course of the estrous cycle; this allows waves of follicles to grow and mature, ovulation to be triggered precisely, and provides the hormonal support necessary for pregnancy.

Single-cell RNA sequencing (scRNAseq) has been used in a variety of tissues to obtain an in-depth understanding of gene expression and cellular diversity. In the ovary, this technique has allowed us, and others, to explore various physiological processes during early ovarian development and ovarian aging (*Zhao et al., 2020*; *Stévant et al., 2019*; *Wagner et al., 2020*; *Niu and Spradling, 2020*; *Jevitt et al., 2020*; *Man et al., 2020*; *Meinsohn et al., 2021*; *Fan et al., 2019*; *Wang et al., 2020*). For example, Fan et al. cataloged the transcriptomic changes that occur during follicular development and regression and mapped the cell types of the human ovary using surgical specimens (*Fan et al., 2019*). A primate model has been used to investigate changes in cell types and states that occur in the ovary with aging (*Wang et al., 2020*). Zhao et al. looked at the formation of the follicle during early embryonic ovarian development to discern the relationship of oocytes to their support cells in formation of follicles (*Zhao et al., 2020*). We have used scRNAseq to identify inhibitory pathways regulated by anti-Müllerian hormone (AMH) during the first wave of follicular growth in the murine ovary (*Meinsohn et al., 2021*). While all these studies have helped establish a static framework to understand the major cell types in the ovary, they fail to describe the dynamic nature of cell states across the reproductive cycle, known as the estrous cycle. The estrous cycle in mice is analogous to the human menstrual cycle, which both reflect follicle development in the ovary. In mice, this cycle lasts 4–5 days and is composed of four different phases known as proestrus, estrus, metestrus, and diestrus. The murine proestrus is analogous to the human follicular stage and leads to ovulation at estrus. Metestrus and diestrus are analogous to early and late secretory stages of the reproductive cycle in humans, which are orchestrated by production of progesterone by the CL (*Ajayi and Akhigbe, 2020*).

To understand more fully the dynamic effects of cyclic endocrine, autocrine, and paracrine signals on ovarian cell states, we performed high-throughput scRNAseq of ovaries from adult mice across a physiological spectrum of reproductive states. Ovaries were harvested from mice in the four phases of the normal estrous cycle: proestrus, estrus, metestrus, and diestrus. Additionally, ovaries were evaluated from mice that were either lactating or non-lactating 10 days post-partum, and from randomly cycling adult mice to increase the diversity of cell states represented in the dataset. Herein, we (1) describe the previously unrecognized complexity in the ovarian cellular subtypes and their cyclic expression states during the estrous cycle, and (2) identify secreted factors that cycle and thus could represent potential biomarkers for staging.

## Results

### scRNA-seq of adult mouse ovaries across reproductive states

To survey the dynamic transcriptional landscape of ovaries at the single-cell level across a range of physiological reproductive states in sexually mature female mice, we isolated the ovaries (four mice per group) at each stage of estrous cycling (proestrus, estrus, metestrus, and diestrus), post-partum non-lactating (PPNL) (day 10 post-partum, with pups removed on the day they were born), post-partum lactating (day 10 post-partum, actively lactating with pups), and non-monitored adult mice to increase sample diversity and cell counts. Following enzymatic digestion of the ovaries, we generated single-cell suspensions and sorted them by microfluidics using the inDROP methodology (*Klein et al., 2015*), targeting 1500 cells per animal. Resulting libraries were indexed and combined for sequencing (*Figure 1A*).

Following dimensionality reduction and clustering using the Seurat algorithm, we identified multiple clusters which could be combined to represent the major cell categories of the ovary (*Figure 1B*). To assign cell type identity, we used cluster-specific markers which were previously described in other studies or newly identified makers later validated by RNA in situ (*Supplementary file 2*). The largest groups of clusters consisted of granulosa cells (N=17627 cells) and mesenchymal cells of the ovarian stroma (N=10825 cells). Other minor cell types were identified including endothelial cells (N=3501 cells), ovarian surface epithelial cells (N=1088 cells), immune cells (N=1649 cells), and oocytes (N=22 cells), altogether recapitulating all the major cell types of the ovary (*Figure 1— figure supplement 1A*). Oocytes were poorly represented in the dataset due to cell size limitations of

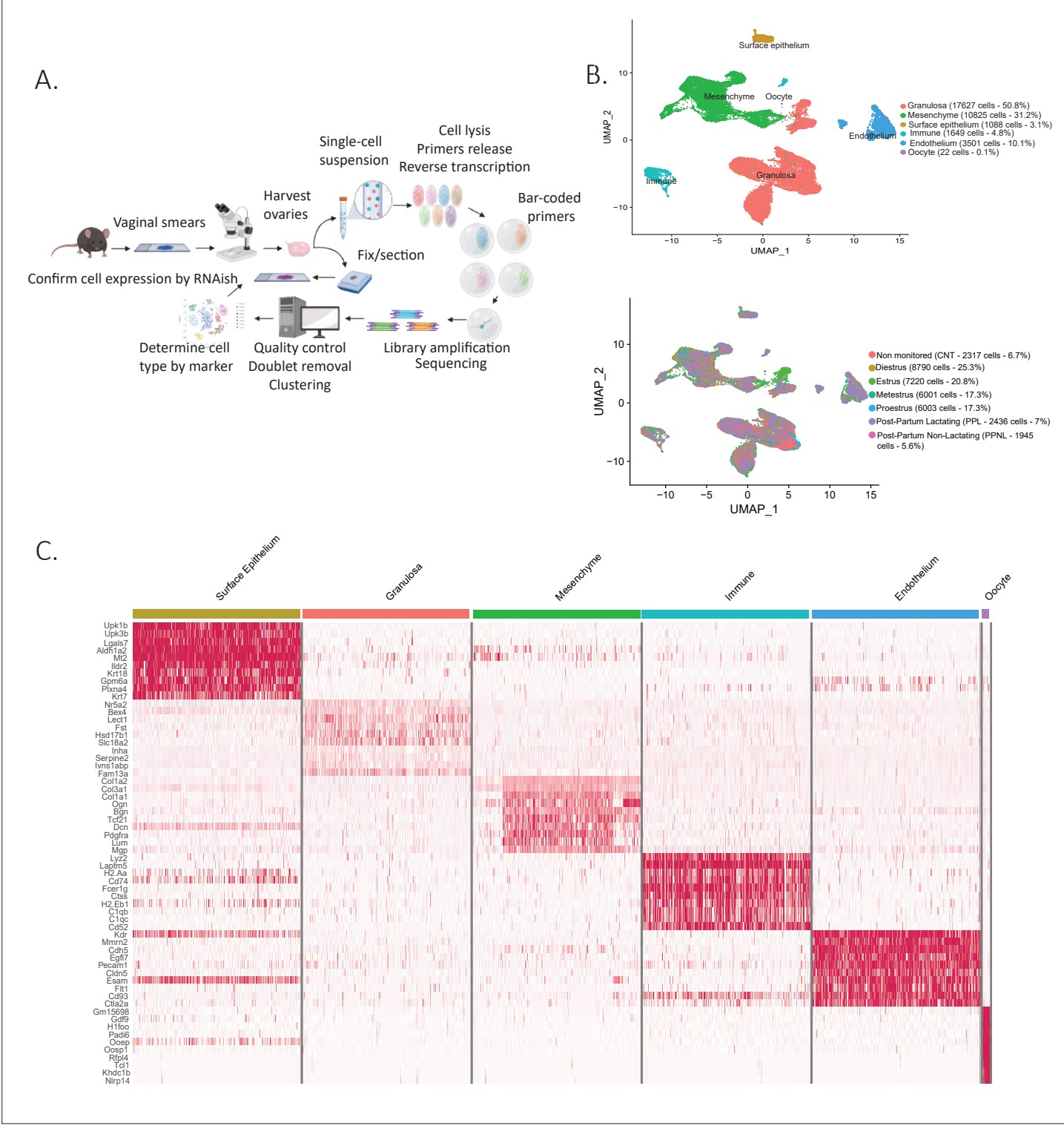

**Figure 1.** Single-cell RNA sequencing of cycling mouse ovaries. (**A**) Schematic of the single-cell sequencing pipeline. (**B**) Uniform manifold approximation and projection (UMAP) plot featuring the different clusters of the ovary and their composition by stage of the estrous cycle, lactating status, or unmonitored. (**C**) Heatmap of the top 10 markers of each cluster by fold change.

The online version of this article includes the following figure supplement(s) for figure 1:

**Figure supplement 1.** Ovarian morphology by reproductive state.

inDROP, likely restricting our sampling to small oocytes of primordial follicles (*Figure 1B*). To characterize more fully the transcriptional signatures of the identified cell types, we evaluated a heatmap of marker gene expression across the major categories of cell types and states (*Figure 1C*). Cells were also classified depending on the stage of the estrous cycle or lactating states in which the ovaries were collected (*Figure 1B*). Morphological differences between the stages of proestrus, estrus, metestrus, diestrus, and also post-partum lactating and non-lactating, were documented in *Figure 1—figure supplement 1B*. The granulosa, mesenchyme, and epithelium clusters were isolated and reanalyzed to identify subclusters.

## Single-cell sequencing reveals heterogeneity within granulosa and mesenchymal cell clusters

### Cellular diversity of mesenchymal cells

The mesenchymal cluster was the second largest cluster identified in our analysis. Based on prior studies and conserved marker expression (*Fan et al., 2019*; *Wang et al., 2020*), we were able to identify subclusters within mesenchymal cells and their relative abundance (percentage) as follows: early theca (16.8%), which formed the theca interna of preantral follicles; steroidogenic theca (13.2%), which formed the theca interna of antral follicles; smooth muscle cells (10.2%), which were part of the theca externa of both antral and preantral follicles; pericytes (6.2%), which surrounded the vasculature; and two interstitial stromal cell clusters, one composed of steroidogenic cells (28.7%) and the other of fibroblast-like cells (24.9%), which together constituted the bulk of the ovarian volume (outside of follicles). These subclusters can be seen in *Figure 2A*, with the top five expressed markers of each subcluster described in the *Figure 2B* heatmap and the top 10 listed in *Supplementary file 3*.

Distinct transcriptional signatures were identified in each of these mesenchymal subclusters (*Figure 2B*); to confirm the presumed identity and histology of these cell types (detailed in *Figure 1—figure supplement 1A*), we validated markers prioritized by highest fold-change expression, highest differential percent expression, and lowest p value (*Figure 2C*).

For the theca interna, the two clusters identified reflected the stage of development of the follicle: early thecal cells could be defined by their expression of hedgehog-interacting protein (*Hhip*) and were histologically associated with preantral follicles. Meanwhile, the steroidogenic theca cells were identified by their expression of cytochrome P450 family 17 subfamily A member 1 (*Cyp17a1*), an essential enzyme for androgen biosynthesis (*Richards et al., 2018*); they were found in antral follicles (*Figure 2C*). The theca externa is a connective tissue rich in extracellular matrix situated on the outermost layer of the follicle (*Figure 1—figure supplement 1A*), containing fibroblasts, macrophages, blood vessels, and abundant smooth muscle cells, which we identified based on expression of microfibril-associated protein 5 (*Mfap5*) by RNA in situ hybridization (*Figure 2C*). To validate the identity and histology of these smooth muscle cells, we performed RNAish/IHC colocalization of *Mfap5* and actin alpha 2 (*Acta2*), another marker of smooth muscle, which confirmed their position within the theca externa. In contrast, *Hhip*, which was expressed in theca interna (both immature and steroidogenic), did not colocalize with Acta2 (*Figure 2—figure supplement 1A-C*). These results suggest Mfap5 labels smooth muscle cells of the theca externa more specifically than Acta2; these cells are thought to perform a contractile function during ovulation (*Young and McNeilly, 2010*).

Lastly, the bulk of the ovarian interstitial stromal space was made up of two closely related cell types which could not be differentiated by specific dichotomous markers but rather were distinguished based on relative expression of ectonucleotide pyrophosphatase/phosphoiestrase 2 (*Enpp2*) (*Figure 2C*). While Enpp2+ cells represented fibroblast-like stromal cell, Enpp2– interstitial cells were enriched for expression of genes such as Patch1 (*Ptch1*), a member of the hedgehog-signaling pathway, an important regulator of ovarian steroidogenesis (*Spicer et al., 2009*), suggesting these represented steroidogenic stromal cells. Indeed, the steroidogenic activity of this stromal cell cluster was further confirmed by its high relative expression of other genes associated with steroidogenesis including cytochrome P450 family 11 subfamily A member 1 (*Cyp11a1*), hydroxy-delta-5-steroid dehydrogenase, 3 beta- and steroid delta-isomerase 1 (*Hsd3b1*), cytochrome P450 family 17 subfamily A member 1 (*Cyp17a1*), steroid 5 alpha-reductase 1 (*Srd5a1*), along with other markers such as potassium two pore domain channel subfamily K member 2 (*Kcnk2*) (*Figure 2—figure supplement 1E, F*). In contrast the fibroblast-like stromal cluster had enriched expression of many extracellular matrix genes such as collagen type I alpha 1 chain (*Col1a1*), collagen type V alpha 1 chain (*Col5a1*), Lumican

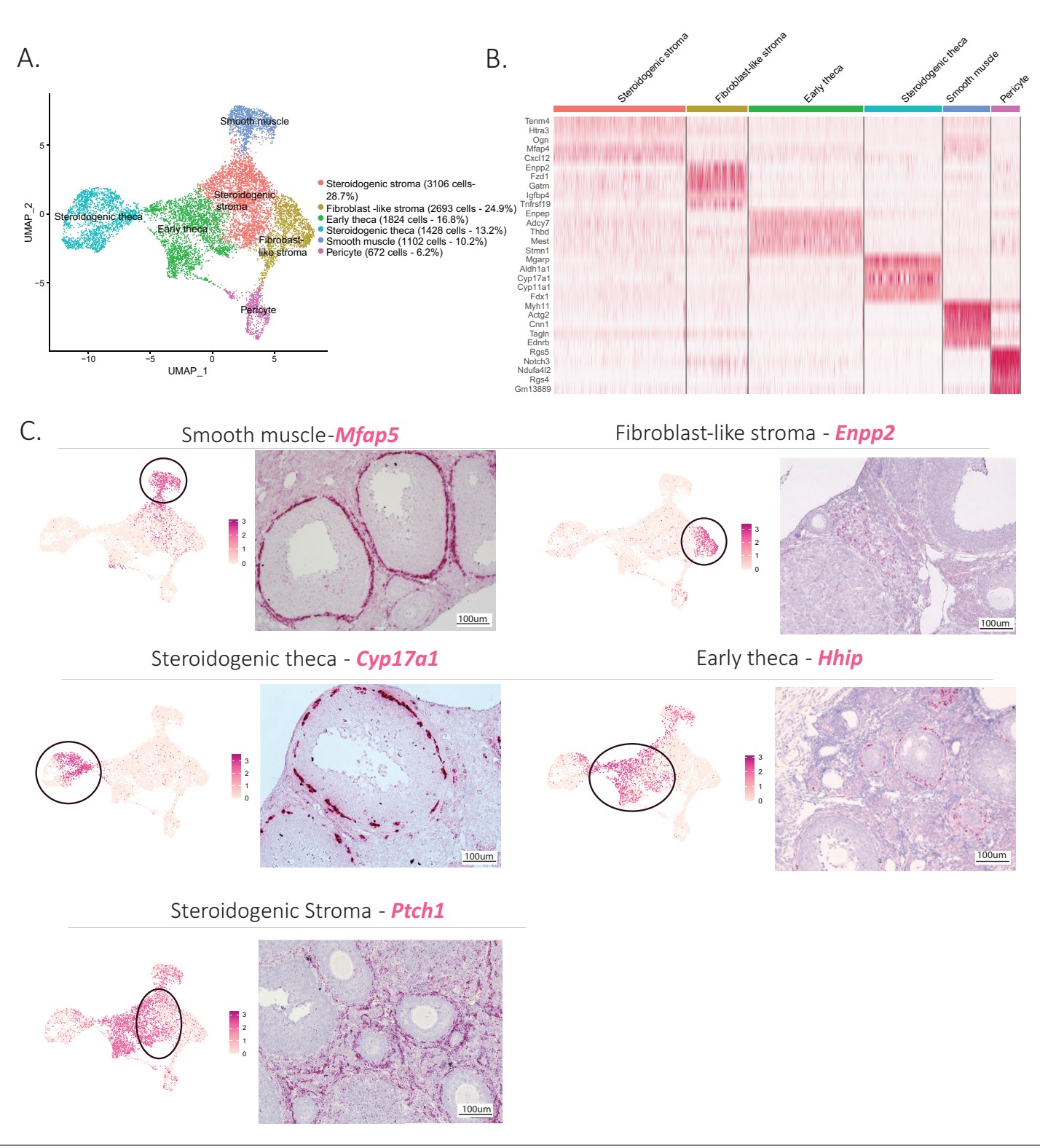

**Figure 2.** Identification of the different cell types of the mesenchyme cluster. (**A**) UMAP plot featuring the different cell subclusters belonging to the mesenchyme cluster. (**B**) Heatmap of the top five markers of each subcluster by fold change. (**C**) Validation of the identity of mesenchyme subcluster by UMAP-plots (cluster of interest circled) and RNA in situ hybridization.

The online version of this article includes the following figure supplement(s) for figure 2:

**Figure supplement 1.** Characterization of mesenchymal cell clusters.

(*Lum*), lysyl oxidase like 1(*Loxl1*) (*Muhl et al., 2020*) along with other known markers of fibroblasts such as C-X-C motif chemokine ligand 14 (*Cxcl14*) (*Lu et al., 2016*) and WT1 transcription factor (*Wt1*) (*He et al., 2008*; *Figure 2—figure supplement 1E, F*).

The identities of above-described mesenchymal clusters matched those of known ovarian stromal cell types based on expression of previously reported markers such as Desmin (Des) for pericytes (*Hughes and Chan-Ling, 2004*), steroidogenic acute regulatory protein (Star) for the steroidogenic theca (*Kiriakidou et al., 1996*), cellular communication network factor 1 (Ccn1) for smooth muscle cells (*Yang et al., 2018b*), receptor activity-modifying protein 2 (Ramp2) for the early theca cells surrounding preantral follicles (*Hatzirodos et al., 2015*), and finally C-X-C motif chemokine ligand 12 (Cxcl12) which we found in both stromal clusters (*Porcile et al., 2005*) as illustrated in *Figure 2— figure supplement 1D*.

## Cellular diversity of granulosa cells

To explore further the cellular heterogeneity within developing follicles (listed in *Figure 1—figure supplement 1A*), we investigated the subclustering of granulosa cells based on their transcriptional profile. Consistent with previous reports, we could distinguish discrete granulosa cell states in follicles based on their stage of development (*Zhao et al., 2020*; *Fan et al., 2019*; *Gallardo et al., 2007*). Granulosa cells could be subdivided into eight main categories: preantral-cumulus (27.3%), antral-mural (21.8%), luteinizing mural (4.8%), atretic (22.6%), mitotic (14.4%), regressing CL (3.7%), and active CL (5.4%) (*Figure 3A*). *Supplementary file 4* lists the top 10 markers for each of these clusters. Distinctive gene expression programs were identified in the granulosa cell subclusters, as visualized in the heatmap (*Figure 3B*), from which we selected potential markers for validation.

Early preantral granulosa cells, and those constituting the cumulus oophorus of antral follicles, could be identified by their shared expression of markers such as potassium channel tetramerization domain (*Kctd14*) (*Figure 3C*), which we had previously shown to be expressed by preantral follicles (*Meinsohn et al., 2021*). In contrast, mural granulosa cells of antral follicles expressed distinct markers (*Supplementary file 4*) such as male-specific transcription in the developing reproductive organs (*Mro*) (*Figure 3C*). Luteinizing mural granulosa cells could be identified by the expression of previously established markers (*Supplementary file 4*) and oxytocin receptor gene (*Oxtr*) which we propose as a highly specific marker for this cell type, a likely target of the surge in oxytocin during estrus (*Ho and Lee, 1992*; *Figure 3C*). Furthermore, we identified two different clusters that we hypothesize represent cell states of the CL, either active or regressing, which both expressed nuclear paraspeckle assembly transcript 1 (*Neat1*), a known marker of CLs (*Nakagawa et al., 2014*). To confirm the active and regressing CL cell states, we investigated the expression of Top2a, a mitotic marker (*Donadeu et al., 2014*), which was enriched in the active CL cluster, and Cdkn1a, a cell cycle exit and senescence marker (*Ock et al., 2020*), which was enriched in the regressing cluster (*Figure 3—figure supplement 1B, C*). Moreover, when examining the composition of clusters depending on the reproductive stage, the regressing CL cluster was found to be composed mostly of cells derived from the Postpartum non lactating (PPNL) samples (*Figure 3—figure supplement 1E*), which overexpressed markers related to CL regression (*Talbott et al., 2017*; *Figure 3—figure supplement 1F*), consistent with a post-partum effect of prolactin. Finally, two relatively abundant granulosa cell states could be identified based on marker expression: mitotic granulosa cells could be found in both preantral and antral follicles and were defined by their expression of *Top2a*, and atretic granulosa cells, which expressed markers consistent with follicular atresia and apoptosis such as phosphoinositide-3-kinase-interacting protein 1 (*Pik3ip1*), nuclear protein 1, transcriptional regulator (*Nupr1*), growth arrest and DNA damage inducible alpha (*Gadd45a*), vesicle amine transport 1 (*Vat1*), transgelin (*Tagln*), and melanocyte-inducing transcription factor (*Mitf*) (*Terenina et al., 2017*; *Figure 3C*, *Figure 3—figure supplement 1A*, *Supplementary file 4*). Furthermore, we propose growth hormone receptor (*Ghr*), which was highly specific to this cluster, as a specific marker of atretic follicles, which warrants further investigation of the role of growth hormone in this process (*Figure 3C*).

## Cellular states in the ovarian surface epithelium

The epithelial cluster was composed of 1088 ovarian surface epithelium (OSE) cells, which could be further subdivided into two clusters (*Figure 4A*): the larger one composed of non-dividing epithelium cells (96%), and a smaller cluster (4%), composed of mitotic epithelium. The latter was characterized

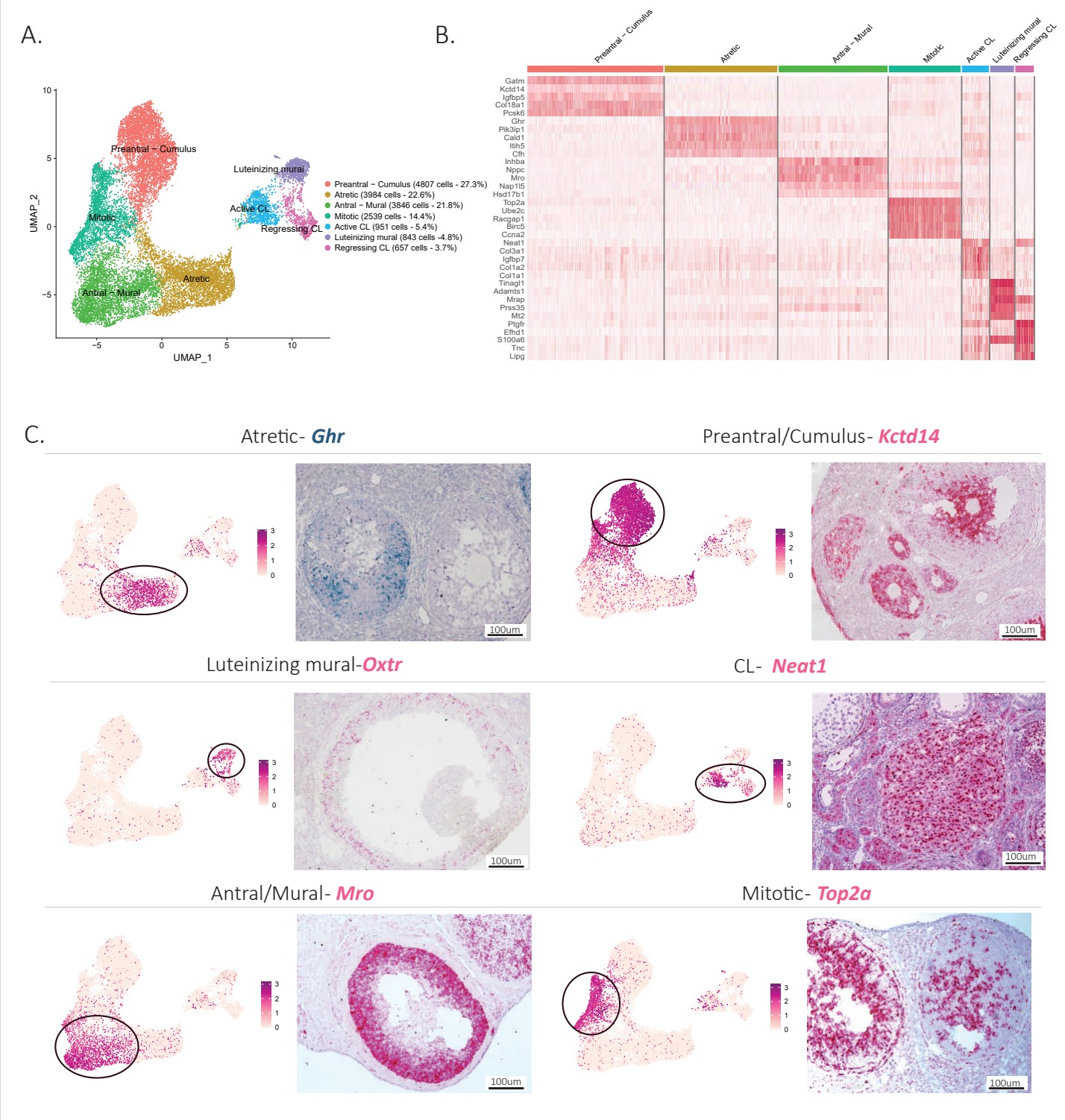

**Figure 3.** Identification of the different cell types in the granulosa cluster. (**A**) UMAP plot featuring the different cell subclusters belonging to the granulosa cluster (specific subcluster circled in each UMAP). (**B**) Heatmap of the top five markers of each cluster by fold change. (**C**) Validation of the identity of granulosa subclusters by UMAP-plots and RNA in situ hybridization.

The online version of this article includes the following figure supplement(s) for figure 3:

**Figure supplement 1.** Characterization of active and regressing corpus luteum clusters.

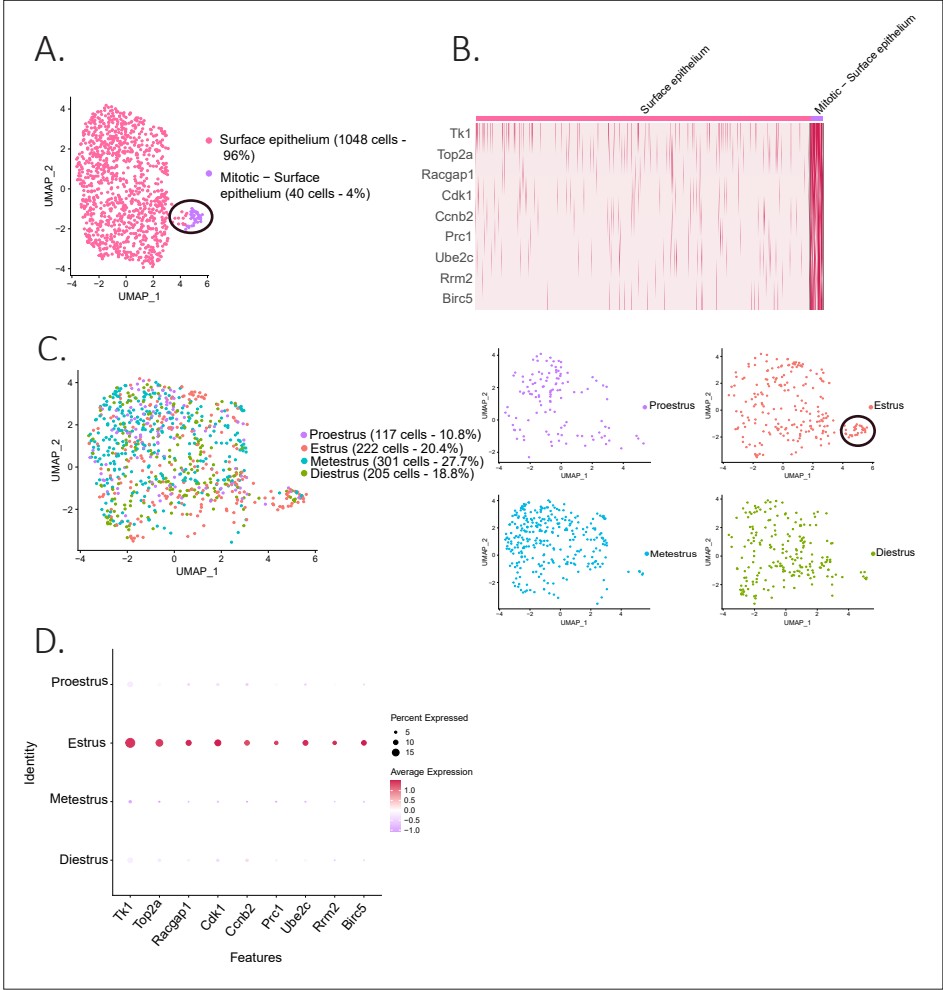

**Figure 4.** Identification of epithelial subclusters. (**A**) UMAP plot of the surface epithelium cluster showing two subclusters: epithelium and mitotic epithelium (circled in black). (**B**) Heatmap of proliferation markers expressed in the proliferating epithelium cluster. (**C**) UMAP plot of the cellular composition of the epithelium subclusters by reproductive state (mitotic subcluster circled in the estrous state). (**D**) Expression of proliferation markers depending on the phase of the estrous cycle.

by proliferation markers such as thymidine kinase 1 (*Tk1*) (*Liu et al., 2019*), Rac GTPase-activating protein 1 (*Racgap1*) (*Yang et al., 2018a*), Top2a, casein kinase 1 (*Ck1*) (*Gao et al., 2021*), protein regulator of cytokinesis 1 (*Prc1*) (*Liang et al., 2019*), ubiquitin-conjugating enzyme E2 C (*Ube2c*) (*Xiong et al., 2019*), and baculoviral IAP repeat containing 5 (*Birc5*) (*Xu et al., 2021*; *Figure 4A and B*). Interestingly, the proliferating subcluster of OSE was almost exclusively composed of cells from the estrous stage (*Figure 4C and D*), consistent with their transient amplification during ovulatory wound closure (*Mara et al., 2020*).

## Granulosa cell transcriptome is most dynamic during the proestrus/ estrus transition

To identify changes in cell states associated with the stages of the estrous cycle, we focused on the granulosa cell subclusters, given the importance of follicular maturation in coordinating this process (illustrated in *Figure 1—figure supplement 1B*). When comparing the composition of granulosa cell subclusters by estrous stage, we found that some clusters were dominated by cells from either the proestrous or estrous samples, particularly the clusters corresponding to 'antral/mural' and 'perio-vulatory' clusters, respectively (*Figure 5A and B*). A volcano plot analysis confirmed that the transition between these two stages was characterized by 24 significantly upregulated and 10 significantly

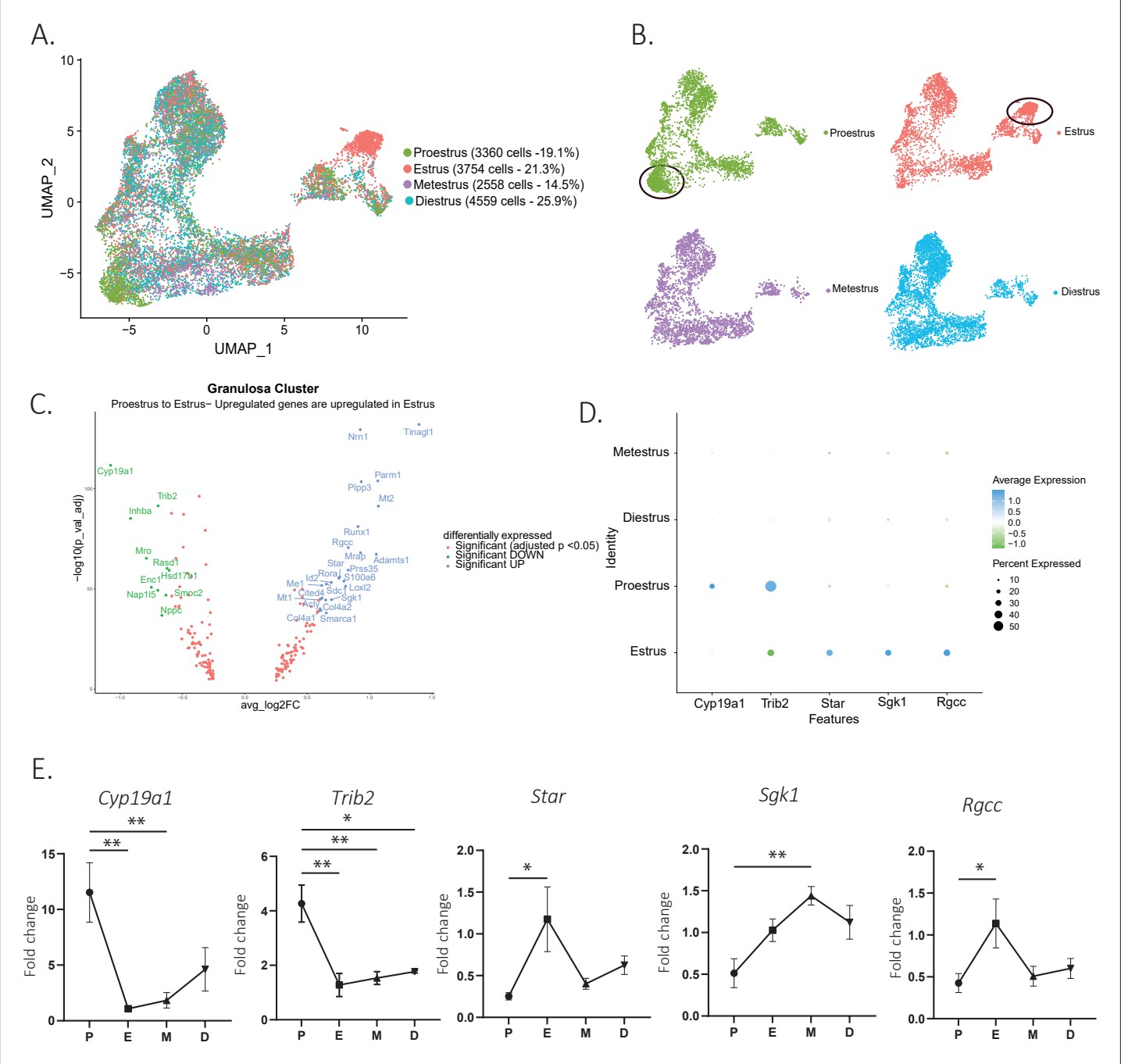

**Figure 5.** Gene expression in granulosa cells by estrous stage. (**A**) UMAP plot featuring estrous cycle stages in the granulosa cell cluster. (**B**) UMAP plots featuring each of the estrous cycle phases individually (proestrus and estrus enriched subclusters circled in black). (**C**) Volcano plot of genes differentially expressed between proestrus and estrous stages. (D) DotPlot of differentially expressed markers between proestrus and estrus. (**E**) Quantitative PCR (qPCR) validation of differentially expressed genes involved in extracellular matrix remodeling and steroidogenesis markers (n=5 per group, mean ± SEM, *p<0.05, **p<0.01, ***p<0.005, and ****p<0.001).

The online version of this article includes the following figure supplement(s) for figure 5:

**Figure supplement 1.** Characterization of the granulosa cell transcriptome across the estrous cycle.

downregulated markers (*Figure 5C*), which together with the transition from estrus to metestrus represents the largest change in gene expression. In contrast, few genes were found to change significantly during the transition from metestrus to diestrus, or diestrus to proestrus (*Figure 5—figure supplement 1A*). Gene ontology analysis revealed that the most significantly differentially regulated pathways between the proestrous and estrous phases were related to ovarian matrix remodeling and steroidogenesis and hormones production (*Figure 5—figure supplement 1B*). To validate the genes with significant changes in expression identified within the single-cell sequencing dataset, we performed quantitative PCR (qPCR) on whole-ovary samples at the proestrus to estrus transition, including the steroid biosynthesis markers cytochrome P450 family 19 subfamily A member 1 (*Cyp19a1*, p=0.0029, proestrus to estrus), *Star* protein (p=0.0187, proestrus to estrus), serum- and glucocorticoid-inducible kinase-1 (*Sgk1*, p=0.0056, proestrus to metestrus), as well as matrix remodeling genes such as regulator of cell cycle (*Rgcc*, p=0.0441, proestrus to estrus), tribbles pseudokinase 2 (*Trib2*, p=0.0023, proestrus to estrus) (*Figure 5D and E*), and immediate early genes, fos proto-oncogene (*Fos*), jun proto-oncogene (*Jun*, p=0.0022, proestrus to estrus), jun proto-oncogene B (*Junb*, p=0.0069, proestrus to diestrus), and early growth response 1 (*Egr1*, p=0.0504 estrus to diestrus), which represent a family of genes thought to be involved in wound repair, a sequela of ovulation (*Florin et al., 2006*; *Wu et al., 2009*; *Martin and Nobes, 1992*; *Yue et al., 2020*; *Figure 5—figure supplement 1C*). Transcriptional gene expression changes were found to be concordant between the scRNAseq data and whole-ovary transcripts quantified by qPCR.

## Identification and validation of secreted biomarkers varying throughout the estrous cycle

To identify new biomarkers that vary as a function of the estrous cycle and that could be used for staging in reproductive medicine, we screened for differentially expressed secreted factors (DAVID Bioinformatics Resources) (*Sherman et al., 2022*; *Huang et al., 2009*), which would therefore be potentially measurable in the blood. Furthermore, to ensure specificity, we prioritized genes expressed specifically in the granulosa or ovarian mesenchymal clusters and not highly expressed in other tissues based on their GTEX profile (*GTEx Consortium, 2013*; *Supplementary file 5*). As a primary screen, we first validated our ability to detect gene expression changes by estrous stage using whole-ovary qPCR analysis in a separate set of staged mice (N=4 per group). Whole-ovary qPCR successfully detected expression changes of estrous cycle markers such as luteinizing hormone/choriogonadotropin receptor (*Toms et al., 2017*) (*Lhcgr*, p=0.0281 estrus to metestrus) and progesterone receptor (*Pgr*, p=0.0096, proestrus to estrus) (*Kubota et al., 2016*; *Figure 6B*). Using this method, we validated a set of significantly upregulated secreted markers in the proestrous to estrous transition, most prominent of which were natriuretic peptide C (*Nppc*, p=0.0022 proestrus to estrus) and inhibin subunit beta-A (*Inhba*, p=0.0067, proestrus to estrus) (*Figure 6A and B*). Similarly, tubulointerstitial nephritis antigen like 1 (*Tinagl1*) and serine protease 35 (*Prss35*) were secreted markers significantly upregulated in estrus compared to their level of transcription in proestrus in the scRNAseq dataset (*Figure 6A*) and by qPCR (*Tinagl1*, p=0.0081, proestrus to estrus; *Prss35*, p=0.0008, proestrus to estrus) (*Figure 6B*). In situ RNA hybridization showed that, as expected, these markers were mostly expressed in mural granulosa cells of antral follicles, while *Nppc* was expressed in both mural and cumulus cells (*Figure 6C*).

To evaluate the feasibility of measuring the secreted PRSS35, NPPC, TINAGL1, and activin A proteins in the serum for staging, we performed ELISAs in mice at each stage of the estrous cycle (*Figure 6D*). We found that the activin A concentration in the serum was significantly increased between the diestrous and proestrous stages (p=0.0312) and peaked at the proestrous stage (*Figure 6D*). The *Inhba* transcript, which encodes for the activin and inhibin beta-A subunit, had a similar temporal expression profile (*Figure 6B*). Circulating PRSS35 levels were lowest at the metestrous stage and were significantly increased during the transition to diestrus (p=0.0009) and remained significantly elevated until the proestrus (*Figure 6D*). In contrast, the Prss35 transcript was significantly induced earlier at estrus (*Figure 6B*). The serum concentrations of TINAGL1, which was lowest at the diestrous and proestrous stages, was significantly increased during the transition between proestrus and metestrus, peaking in estrus (p=0.0142) (*Figure 6D*). This temporal pattern of expression was recapitulated at the transcriptional level by qPCR and scRNAseq (*Figure 6A and B*). Finally, we observed a trend for serum protein concentrations of NPPC to be lowest at the proestrous and estrous stages

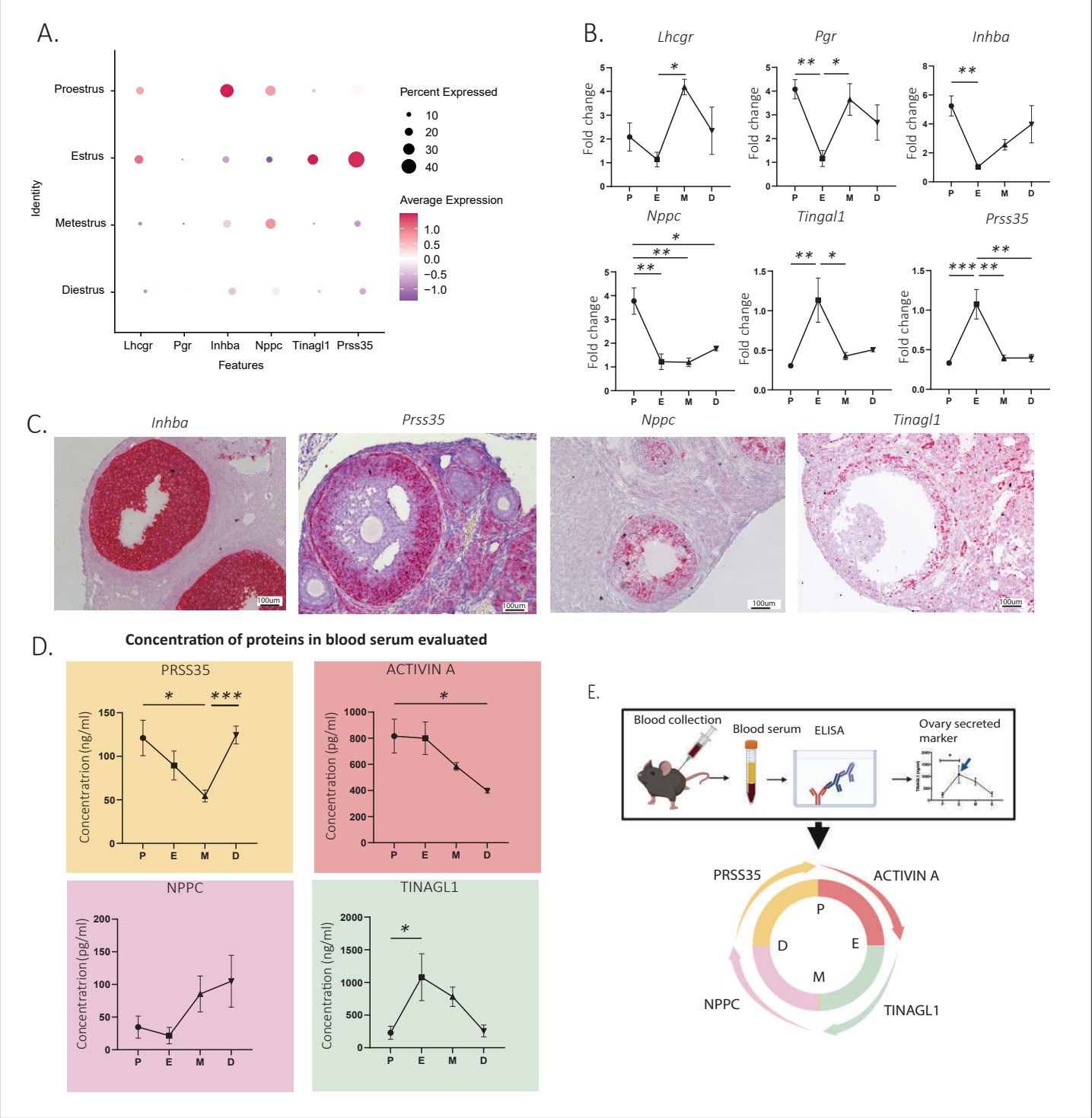

**Figure 6.** Identification and validation of new secreted estrous staging markers. (**A**) Expression of granulosa cell transcripts varying by estrous cycle stage. (**B**) Validation of significantly up- and downregulated transcripts of secreted estrous staging markers by qPCR (n=5 per group, mean ± SEM, *p<0.05, **p<0.01, and ***p<0.005). (**C**) Localization of estrous staging markers by in situ hybridization (RNAscope) in ovarian sections. (**D**) Quantification of circulating estrous staging markers proteins in the blood by enzyme-Linked Immunosorbent assay (ELISA) (n=5 per group, mean ± SEM, *p<0.05, and ***p<0.005). (**E**) Summary of the timing of expression of estrous staging markers in the blood.

and increase during the metestrous and diestrous stages (*Figure 6D*), although the differences were not statistically significant (p=0.0889, estrus to metestrus). Importantly, these data provide a proof of concept that four markers could be used to monitor estrous cycle progression when measured in conjunction in the blood (*Figure 6E*).

## Discussion

scRNAseq has been used to catalog the transcriptomes of a variety of tissues in several species, across different physiological states (*Hwang et al., 2018*). Herein, we used scRNAseq to survey the cellular diversity and the dynamic cell states of the mouse ovary across the estrous cycle and other reproductive states such as post-partum lactating (PPL) and post-partum non-lactating (PPNL).

The most significant changes in composition and cell states were identified in granulosa cells, particularly as they cycled through the estrous stages, reflective of their important role in cyclic follicular maturation and hormone production. Early preantral follicle numbers are thought to be relatively stable across the estrous cycle (*Deb et al., 2013*), given that they are largely unresponsive to gonadotropins (*Richards, 1980*), in contrast to antral follicles, whose numbers and size are more variable (*Deb et al., 2013*). Indeed, while subclusters such as 'preantral granulosa cells' were equally represented in samples from proestrus, metestrus, and diestrus, others, such as the 'luteinizing mural' cluster, were dominated by cells derived from one stage (in this case 'estrus'). Genes enriched in this cluster had been previously reported to be involved in the ovulatory process and regulated by the luteinizing hormone (LH) surge, including markers of terminal differentiation and steroidogenesis such as *Smarca1* (*Lazzaro et al., 2006*), *Cyp11a1* (*Irving-Rodgers et al., 2009*), metallothionein 1 (*Mt1*), and metallothionein 2 (*Mt2*) (*Wang et al., 2018*; *Supplementary file 4*). Other genes enriched in this subcluster include Prss35 (*Wahlberg et al., 2008*) and Adamts1 (*Lussier et al., 2017*; *Sayasith et al., 2013*), which had previously been identified as playing a role in the follicular rupture necessary for ovulation.

Interestingly, we found that granulosa cells of preantral follicles and cumulus cells of antral follicles clustered together and shared markers that distinguished them from mural granulosa cells. For example, Kctd14, a member of the potassium channel tetramerisation domain-containing family, was expressed in granulosa cells during the initial growth of early preantral follicles, but also specifically expressed only in cumulus cells, but not mural granulosa cells, of larger antral follicles. Intriguingly, we have previously shown that AMH (anti- Müllerian hormone, a.k.a Müllerian Inhibiting Substance), which is specifically expressed by cumulus cells (*Diaz et al., 2007*) in antral follicles, regulates KCTD14 expression in preantral follicles (*Meinsohn et al., 2021*). This conservation of cellular state and marker expression from preantral granulosa cells to cumulus cells of antral follicles suggests a continuous lineage, potentially defined and maintained by the close interaction with the oocyte (*Diaz et al., 2007*). This interpretation is consistent with the presence of a differentiation fork in the granulosa cell lineage during antrum formation, which would give rise to a distinct mural granulosa cell fate poised to respond to the LH surge. Indeed, the periovulatory granulosa cell state was identified based on its expression of genes regulated by LH such as Smarca1 (*Lazzaro et al., 2006*), and we propose Oxtr as a specific marker for these cells (*Figure 3—figure supplement 1D*). Oxtr expression was found only in the mural cells of large Graafian follicles, suggesting it indeed corresponds to an LH-stimulated mural granulosa cell state.

After ovulation, these LH-stimulated mural granulosa cells, along with the steroidogenic theca cells, terminally differentiate into luteal cells and form the corpus luteum (CL). The CL is a transient structure with highly active steroid biosynthesis, providing the progesterone to maintain pregnancy (*Duncan, 2021*). In absence of implantation, the CL degenerates (*Noguchi et al., 2017*). We found this progression of the CL to be recapitulated at the transcriptional level, leading to two luteal subclusters: active CL and regressing CL. While the active CL was characterized by expression of proliferation markers (Top2a) in addition to steroidogenic enzymes, the regressing CL expressed the cell cycle inhibitor and senescence maker Cdkn1a, along with luteolysis markers such as syndecan 4 (Sdc4), claudin domain containing 1 (Cldnd1), and BTG anti-proliferation factor 1 (Btg1) (*Talbott et al., 2017*; *Zhu et al., 2013*). The distinct expression signatures observed in these two clusters may provide insights into the molecular basis of luteolysis and warrant further investigation.

The mesenchymal cluster was also surprisingly complex and variable across the estrous cycle, reflecting stromal remodeling and other physiological functions supporting follicle growth, steroid

hormone production, and ovulation. For example, during follicle maturation, the ovarian stromal cells adjacent to the developing follicle differentiates into the theca, which is ultimately responsible for steroid hormone biosynthesis and therefore underlies the cyclic hormone production of the ovary (*Ryan and Petro, 1966*). Herein, we identified two thecal clusters, designated as early and steroidogenic theca. Early theca was defined by markers such as Hhip (*Richards et al., 2018*; *Hummitzsch et al., 2019*), mesoderm-specific transcript (Mest) (*Fan et al., 2019*), and patched 1 (Ptch1) (*Fan et al., 2019*; *Richards et al., 2018*) and given its association with small follicles, presumed to be immature and the precursor to steroidogenic theca. As the follicle matures and the antrum forms, this layer becomes more vascularized and differentiates into theca interna, which is steroidogenic. This steroidogenic theca cluster was readily identifiable through its expression of steroidogenic enzymes such as Hsd3b1, Cyp17a1, Cyp11a1, and also well-established markers such as ferredoxin-1 (Fdx1) and prolactin receptor (Prlr) (*Fan et al., 2019*; *Grosdemouge et al., 2003*). Interestingly, we confirmed the presence of steroidogenic interstitial stromal cells which also likely contribute to sex steroid production in the ovary. Indeed, such cells likely represent the precursors of the theca interna (*Sheng et al., 2022*; *Kinnear et al., 2020*). Smooth muscle cells, which are part of the theca externa, were identified by their expression of structural proteins such as Mfap5, myosin heavy chain 11 (Myh11), Tagln, and smooth muscle actin (Sma or Acta2) (*Zhao et al., 2020*). In contrast to mice, human smooth muscle cells are thought to express high levels of collagen (*Fan et al., 2019*), which we did not observe here. Another species difference between mice and humans was the expression of aldehyde dehydrogenase 1 family member A1 (Aldh1a1), which we found primarily in the steroidogenic theca cluster, while it is presumably enriched in the theca externa in humans (*Fan et al., 2019*).

Ovulation is associated with a dramatic remodeling of the ovary, including the subsequent ovulatory wound repair. We identified fibroblast-like cells in the ovarian stroma expressing many of the extracellular matrix protein known to play a role in these processes (*Mara et al., 2020*; *Duffy et al., 2019*). Another important player in ovulatory would repair is the ovarian surface epithelium (OSE), a simple mesothelial cell layer that covers the surface of the ovary and must dynamically expand to cover wound (*Hartanti et al., 2019*; *Xu et al., 2011*). The OSE cluster could be identified based on well-established markers such as keratin (Krt) 7, 8, and 18 (*Kenngott et al., 2014*) and was represented by only 3% of all cells in our dataset, which could be further subdivided in proliferative and non-proliferative states. As expected from their function in ovulatory wound repair, dividing OSE was enriched during estrus. Furthermore, genes associated with wound healing such as galectin 1 (*Lgals1*) (*Lin et al., 2015*) were also significantly upregulated in estrus. Similarly, the expression of the immediate-early genes Fos, Jun, Junb, and Egr1 was variable during the estrous cycle, following a common pattern of strong downregulation at estrous compared to the other stages, consistent with their temporal expression during the repair of other tissues such as the cornea (*Okada et al., 1996*).

Finally, to take advantage of this rich dataset, we sought to identify secreted markers which varied in abundance during the estrous cycle and could thus be used as staging biomarkers in assisted reproduction. We identified and prioritized four secreted biomarkers, expressed in mouse but also human ovaries, which varied significantly during different transitions of the estrous cycle, namely *Inhba* (*Wijayarathna and de Kretser, 2016*), *Prss35* (*Wahlberg et al., 2008*; *Li et al., 2015*), *Nppc* (*Zhang et al., 2010*; *Xi et al., 2021*), and *Tinagl1* (*Akaiwa et al., 2020*; *Kim et al., 2010*).

Activin A is a secreted protein homodimer translated from the *Inhba* transcript that is a crucial modulator of diverse ovarian functions including pituitary feedback, and whose expression level depends highly on the stage of the estrous cycle (*Chang and Leung, 2018*). Quantification of activin A protein in the serum by ELISA revealed elevated levels in the blood during both proestrus and estrus, which is consistent with studies of other species such as ewes (*OConnell et al., 2016*). Importantly, the protein product of *Inhba*, the inhibin beta-A subunit, can be incorporated into other protein dimers, such as activin BA, and inhibin A, which were not measured in this study and may also represent cycling biomarkers.

The serine protease 35 transcript was expressed in the theca layers of preantral follicles and induced in granulosa cells of preovulatory follicles and all stages of the corpora lutea, peaking at the estrous stage according to qPCR, leading us to speculate that it may be involved tissue remodeling during ovulation and CL formation (*Wahlberg et al., 2008*). In contrast, the PRSS35 protein levels were highest in the diestrus and proestrus stages as determined by ELISA, suggesting other tissue

sources of PRSS35 or an offset in peak protein levels due to delays in accumulation of the protein in the circulation.

The natriuretic peptide precursor C (NPPC) protein is a peptide hormone encoded by the *Nppc* gene. *Nppc* has been reported to be expressed by mural granulosa cells, while its receptor Npr2 is expressed by cumulus cells (*Zhang et al., 2010*). The pair acts on developing follicles by increasing the production of intracellular cyclic guanosine monophosphate and maintains oocyte meiotic arrest during maturation. Upon downregulation of this pathway, the oocyte can escape meiotic arrest and ovulate (*Celik et al., 2019*). This close relationship with the ovulatory process makes *Nppc* an attractive marker to predict ovulation. Herein, qPCR analysis revealed that *Nppc* was highest in the ovary at proestrus and was quickly and significantly downregulated at estrous, probably in response to the increased levels of LH which in turn inhibit the Nppc/Npr2 system (*Celik et al., 2015*). In contrast, there was a trend for the circulating NPCC peptide to be highest in metestrus and diestrus, albeit not in a statistically significant way.

Finally, we evaluated the level of transcription and protein expression of the matricellular factor Tinagl1. We found both the *Tinagl1* transcript and the circulating TINAGL1 protein in the blood to be highest during estrous, thus coinciding with ovulation, with a pattern of expression consistent with expression by mural granulosa cells of antral follicles. While the role of TINAGL1 in the ovary has not been extensively investigated, it has been associated with delayed ovarian collagen deposition and increased ovulation in aging Tinagl1 knock-out mice (*Akaiwa et al., 2020*).

Those four potential cyclic biomarkers, activin A, PRSS35, NPPC, and TINAGL1, provide a proof of concept that a deeper understanding of transcriptional changes at the single-cell level may translate into useful applications in assisted reproduction. It will be of interest to follow up the findings of cyclic expression of these four markers, particularly in combination as an index, for the purpose of staging and predicting ovulation timing in humans and other species .

In summary, this study outlines the dynamic transcriptome of murine ovaries at the single-cell level and across the estrous cycle and other reproductive states, and extends our understanding of the diversity of cell types in the adult ovary. We identified herein novel biomarkers of the estrous cycle that can be readily measured in the blood and may have utility in predicting staging for assisted reproduction. This rich dataset and extensive validation of new molecular markers of cell types of the ovary will provide a hypothesis-generating framework of dynamic cell states across the cycle with which to elucidate the complex cellular interactions that are required for ovarian homeostasis.

## Materials and methods

### Key resources table

| Reagent type (species) or resource | Designation | Source or reference | Identifiers | Additional information |
|---|---|---|---|---|
| Genetic reagent (*Mus musculus*) | C57BL/6-Tg(UBC-GFP)30Scha/J | Jackson Laboratory | stock #004353 | |
| Antibody | Smooth muscle alpha action (SMA) (Rabbit polyclonal) | Abcam | #5694 | Dilution: 1:300 |
| Commercial assay or kit | ACTIVIN A commercial ELISA | RnD systems | #DAC00B | |
| Commercial assay or kit | NPPC commercial ELISA | Novus Bio | #NBP2-75790 | |
| Commercial assay or kit | Tinagl1 commercial ELISA | LS-Bio | #LS-F49684 | |
| Commercial assay or kit | PRSS35 commercial ELISA | Mybiosource | #MBS9717242 | |
| Commercial assay or kit | RNA scope 2.5 HD Duplex detection kit | ACD bio | #322500 | |
| Commercial assay or kit | RNA scope 2.5 HD red detection kit | ACD bio | #322360 | |
| Commercial assay or kit | The target retrieval and protease plus reagents | ACD bio | #322330 | |
| Other | Cdkn1a (*M. musculus*) NM_007669.4 | ACD bio | # 408551 | RNAscope probe |
| Other | Cxcl14 (*M. musculus*) NM_019568.2 | ACD bio | #459741 | RNAscope probe |

*Continued on next page*

*Continued*

| Reagent type (species) or resource | Designation | Source or reference | Identifiers | Additional information |
|---|---|---|---|---|
| Other | Cyp11a1 (*M. musculus*) NM_019779.4 | ACD bio | #809181 | RNAscope probe |
| Other | Cyp17a1 (*M. musculus*) NM_007809.3 | ACD bio | #522611 | RNAscope probe |
| Other | Ghr (*M. musculus*) NM_010284.3 | ACD bio | #464951 | RNAscope probe |
| Other | Hhip (*M. musculus*) NM_020259.4 | ACD bio | #448441 | RNAscope probe |
| Other | Inhba (*M. musculus*) NM_008380.1 | ACD bio | # 455871 | RNAscope probe |
| Other | Kcnk2 (*M. musculus*) NM_001159850.1 | ACD bio | #440421 | RNAscope probe |
| Other | Kctd14 (*M. musculus*) NM_001136235.1 | ACD bio | #517811 | RNAscope probe |
| Other | Mfap5 (*M. musculus*) NM_015776.2 | ACD bio | #490211 | RNAscope probe |
| Other | Mro (*M. musculus*) NM_001305882.1 | ACD bio | #491181 | RNAscope probe |
| Other | Neat1 (*M. musculus*) NR_003513.2 | ACD bio | #440351 | RNAscope probe |
| Other | Nppc (*M. musculus*) NM_010933.5 | ACD bio | # 493291 | RNAscope probe |
| Other | Onecut2 (*M. musculus*) NM_194268.2 | ACD bio | #520541 | RNAscope probe |
| Other | Oxtr (*M. musculus*) NM_001081147.1 | ACD bio | #412171 | RNAscope probe |
| Other | Prss35 (*M. musculus*) NM_178738.3 | ACD bio | #492611 | RNAscope probe |
| Other | Runx1 (*M. musculus*) NM_001111021.1 | ACD bio | #406671 | RNAscope probe |
| Other | Tinagl1 (*M. musculus*) NM_001168333 | ACD bio | #312621 | RNAscope probe |
| Other | Top2a (*M. musculus*) NM_011623.2 | ACD bio | # 491221 | RNAscope probe |
| Other | Wt1 (*M. musculus*) NM_144783.2 | ACD bio | #432711 | RNAscope probe |
| Software and algorithm | R version 4.1.3 | R Project for Statistical Computing | https://scicrunch.org/resolver/SCR_001905 | |
| Software and algorithm | Seurat package 4.1.0 | R toolkit for single-cell genomics | https://satijalab.org/seurat/articles/install.html | |
| Software and algorithm | BZ-X800 analysis software | Keyence | https://www.keyence.com/landing/microscope/lp_fluorescence.jsp | |
| Software and algorithm | GraphPad Prism, version 9.2.0 | Graphpad | | |

## Mice

Animal experiments were carried out in 6–8 weeks old C57BL/6 mice obtained from Charles River Laboratory, approved by the National Institute of Health and Harvard Medical School Institutional Animal Care and Use Committee, and performed in accordance with experimental protocols 2009N000033 and 2014N000275 approved by the Massachusetts General Hospital Institutional Animal Care and Use Committee.

For the analysis of transcriptional changes in ovaries of cycling mice, animals were housed in standard conditions (12/12 hr light/dark non-inverting cycle with food and water ad libitum) in groups of five females with added bedding from a cage that previously housed an adult male mouse to encourage cycling. Estrous stage was determined by observation of the vaginal opening and by vaginal swabs done at the same time daily, as previously described (*Kano et al., 2017*). Each mouse was monitored for a minimum of 2 weeks to ensure its cyclicity. Four mice were sacrificed in each of the four phases of the estrous cycle and labeled as being from experimental batch 'cycling'. An additional eight mice were included in the analysis and labeled as being from experimental batch 'lactating'. Four of these mice were lactating at day 10 post-partum, and four were 10 days post-partum with pups removed at delivery. Four additional mice were not monitored for cycling and included to increase sample size and diversity.

Additional mice were monitored throughout the estrous cycle to collect ovaries at each stage (groups of N=5 for proestrus, estrus, metestrus, and diestrus) for gene validation. Paired ovaries were collected from each staged mouse: one was used to extract mRNA for qPCR, while the other was fixed in 4% paraformaldehyde for RNAish (RNAscope) or immunohistochemistry to validate gene expression.

## Superovulation

To stimulate superovulation, mature female mice (6–9 weeks C57BL/6) were injected intraperitoneally (IP) with 5 IU of pregnant mare serum gonadotropin (PMSG; Calbiochem, San Diego, CA, USA), followed 48 hr later by 5 IU of human chorionic gonadotropin (hCG; Millipore Sigma, St. Louis, MO, USA). The mice were euthanized 8 hr after hCG treatment and ovaries harvested.

## Staging of estrous cycle by vaginal cytology

As previously described (*Kano et al., 2017*; *Byers et al., 2012*), staging of mice was performed using a wet cotton swab, introduced into the vaginal orifice then smeared onto a glass slide which was air-dried, stained with Giemsa, and scored for cytology by two independent observers. Briefly, proestrus was determined if the smear showed a preponderance of nucleated epithelial cells as well as leukocytes. Estrous was marked by an abundance of cornified epithelial cells, while metestrous smears contained a mixture of cornified epithelial cells and leukocytes. Finally, diestrus was characterized by abundant leukocytes with low numbers of cornified epithelium or nucleated epithelial cells.

## Generation of single-cell suspension

Single-cell suspension from mouse ovaries was obtained as previously described with uterine enzymatic dissociation (*Saatcioglu et al., 2019*). Briefly, ovaries were incubated for 30 min at 34°C in dissociation medium (82 mM $Na_2SO_4$, 30 mM $K_2SO_4$, 10 mM Glucose, 10 mM HEPES, and 5 mM $MgCL_2$, pH 7.4) containing 15 mg of Protease XXIII (Worthington), 100 U Papain, with 5 mm L-Cysteine, 2.5 mM EDTA (Worthington), and 1333 U of DNase 1 (Worthington). The reaction was then stopped in cold medium, and samples were mechanically dissociated, filtrated, and spun down three times before being resuspended to a concentration of 150,000 cells/mL in 20% Optiprep (Sigma) for inDrop sorting.

## Single-cell RNA sequencing (inDrop)

Fluidic sorting was performed using the inDrop platform at the Single-Cell Core facility at Harvard Medical School as previously described (*Klein et al., 2015*; *Macosko et al., 2015*). We generated libraries of approximately 1500 cells per animal which were sequenced using the NextSeq500 (Illumina) platform. Transcripts were processed according to a previously published pipeline *Klein et al., 2015* used to build a custom transcriptome from the Ensemble GRCm38 genome and GRCm38.84 annotation using Bowtie 1.1.1. Unique molecular identifiers (UMIs) were used to reference sequence reads back to individual captured molecules, referred to as UMIFM counts. All steps of the pipeline were run using default parameters unless explicitly specified.

## scRNAseq data analysis

### Data processing

The initial Seurat object was created using thresholds to identify putative cells (unique cell barcodes) with the following parameters: 1000–20,000 UMIs, 500–5000 genes, and less than 15% mitochondrial genes. The final merged dataset contained ~70,000 cells which were clustered based on expression of marker genes. These were further processed in several ways to exclude low-quality data and potential doublets. Visualization of single-cell data was performed using a non-linear dimensionality-reduction technique, uniform manifold approximation and projection. Markers for each level of cluster were identified using MAST in Seurat (R version 4.1.3 - Seurat version 4.1.0). Following identification of the main clusters (granulosa, mesenchyme, endothelium, immune, epithelium, and oocyte), we reanalyzed each cluster population to perform subclustering. Briefly, the granulosa, mesenchyme, and epithelium clusters were extracted from the integrated dataset by the subset function. The isolated cluster

was then divided into several subclusters following normalization, scale, principal component analysis (PCA), and dimensionality reductions as previously described (*Niu and Spradling, 2020*).

### Volcano plots
Highly differentially expressed genes between different estrous cycles were identified using the function FindMarkers in Seurat. Volcano plots were generated using ggplot2 package in R.

### Pathway enrichment analysis
Differentially expressed genes with at least twofold changes between contiguous estrous stages were used as input for gene ontology enrichment analysis by clusterProfiler. Enrichplot package was used for visualization. Biological process subontology was chosen for this analysis.

### Principal component analysis
PCA was used to identify common patterns of gene expression across stages of the cycle. For each Level 0 cluster object, cycling cells were extracted, and genes that were expressed in more than 5% of cells were identified. The expression of these genes in the cycling cells were scaled (set to mean zero, SD 1) and averaged across each of the four cycle stages. PCA was run (prcomp) on the average scaled expression data.

## In situ hybridization and immunohistochemistry
In situ hybridizations were performed using ACDBio kits as per manufacturer's protocol, as previously described (*Saatcioglu et al., 2019*). Briefly, RNAish was developed using the RNAscope 2.5 HD Reagent Kit (RED and Duplex, ACD Bio). Following deparaffinization in xylene, dehydration, peroxidase blocking, and heat-induced epitope retrieval by the target retrieval and protease plus reagents (ACD bio), tissue sections were hybridized with probes for the target genes (see Key resources table for accession number and catalog number of each gene) in the HybEZ hybridization oven (ACD Bio) for 2 hr at 40°C. The slides were then processed for standard signal amplification steps and chromogen development. Slides were counterstained in 50% hematoxylin (Dako), air dried, and coverslipped with EcoMount. In addition to cycling and non-cycling mice, superovulated mice were used to validate markers from follicles associated with LH surge response in ovulatory follicles at the estrous stage for more precise timing of collection.

For colocalization of RNAish staining with immunohistochemistry, we first processed the tissue section for RNAscope as described above, including deparaffinization, antigen retrieval, hybridization, and chromogen development. Sections were then blocked in 3% bovine serum albumin in Tris-buffered solution (TBS) for 1 hr. Following three washes with TBS, the sections were incubated with the primary antibody (smooth muscle actin primary antibody; 1:300, Abcam) overnight at 4°C and developed with Dako EnVision + System horseradish peroxidase (HRP). Labeled polymer anti-rabbit was used as the secondary antibody, and the HRP signal was detected using the Dako detection system. Slides were then counterstained in hematoxylin and mounted as described above.

## Reverse transcription-quantitative polymerase chain reaction
Mice were monitored through the estrous cycle and sacrificed at specific stage/timepoints as described above. Ovaries were dissected, and total RNA was extracted using the Qiagen RNA extraction kit (Qiagen). A cDNA library was synthesized from 500 ng total RNA using SuperScript III First-Strand Synthesis System for RT-PCR using manufacturer's instructions with random hexamers (Invitrogen). The primers used for this study are described in *Supplementary file 1*. Expression levels were normalized to the Gapdh transcript using cycle threshold (Ct) values logarithmically transformed by the $2-\Delta Ct$ function.

## ELISA
Blood was collected from mice by facial vein puncture, incubated at room temperature (RT) until spontaneously clotted, centrifuged at 8000 rpm for 5 min to collect the serum layer, and diluted 1/10 in each ELISA kit according to the manufacturing protocol; Mouse CNP/NPPC ELISA kit; Mouse serine

protease inactive 35 (PRSS35) ELISA kit; Mouse TINAGL1 /Lipocalin 7 ELISA kit; and Human/Mouse/ Rat Activin A Quantikine ELISA Kit (see Key resources table).

## Acknowledgements
We thank LiHua Zhang, Bugra Uluyurt, Phoebe A May, Caroline Coletti, and Sarah Mustafa Eisa for technical help. This study was supported by the National Institute for Child Health and Human Development to D.P. (1R01HD102014-01), the Huiying Fellowship (H.D.S. and D.P.), Sudna Gar Fellowship (D.P.), Massachusetts General Hospital Executive Committee on Research (D.P. and P.K.D.), and royalties (P.K.D.) from the use of the MIS ELISA in infertility clinics.

## Additional information

### Funding

| Funder | Grant reference number | Author |
| --- | --- | --- |
| Eunice Kennedy Shriver National Institute of Child Health and Human Development | 1R01HD102014-01 | David Pépin |
| Massachusetts General Hospital | | David Pépin Patricia K Donahoe |
| Huiying Foundation | Huiying Fellowship | Hatice D Saatcioglu David Pépin |

The funders had no role in study design, data collection and interpretation, or the decision to submit the work for publication.

### Author contributions
Mary E Morris, Conceptualization, Data curation, Formal analysis, Funding acquisition, Validation, Investigation, Writing - original draft; Marie-Charlotte Meinsohn, Conceptualization, Data curation, Formal analysis, Validation, Investigation, Methodology, Writing - original draft, Writing - review and editing; Maeva Chauvin, Conceptualization, Data curation, Formal analysis, Investigation, Writing - original draft, Writing - review and editing; Hatice D Saatcioglu, Conceptualization, Data curation, Formal analysis, Investigation, Methodology; Aki Kashiwagi, Data curation, Validation, Investigation, Methodology; Natalie A Sicher, Data curation, Formal analysis, Validation, Investigation; Ngoc Nguyen, Formal analysis, Validation, Investigation; Selena Yuan, Data curation, Validation; Rhian Stavely, Data curation, Formal analysis, Writing - review and editing; Minsuk Hyun, Data curation, Software, Formal analysis, Investigation, Visualization; Patricia K Donahoe, Supervision, Writing - original draft, Writing - review and editing; Bernardo L Sabatini, Conceptualization, Data curation, Software, Formal analysis, Supervision, Funding acquisition, Investigation, Methodology, Writing - original draft, Project administration; David Pépin, Conceptualization, Data curation, Software, Formal analysis, Supervision, Funding acquisition, Investigation, Methodology, Writing - original draft, Project administration, Writing - review and editing

### Author ORCIDs
Marie-Charlotte Meinsohn ⑩ http://orcid.org/0000-0002-1378-4655
Bernardo L Sabatini ⑩ http://orcid.org/0000-0003-0095-9177
David Pépin ⑩ http://orcid.org/0000-0003-2046-6708

### Ethics
Animal experiments were approved by the National Institute of Health and Harvard Medical School Institutional Animal Care and Use Committee, and performed in accordance with experimental protocols 2009N000033 and 2014N000275 approved by the Massachusetts General Hospital Institutional Animal Care and Use Committee.

### Decision letter and Author response
Decision letter https://doi.org/10.7554/eLife.77239.sa1

Author response https://doi.org/10.7554/eLife.77239.sa2

## Additional files

### Supplementary files
- Supplementary file 1. List of primers used for quantitative PCR (qPCR) experiments.
- Supplementary file 2. Top 10 markers expressed in each ovary cluster.
- Supplementary file 3. Top 10 markers from each mesenchyme subclusters.
- Supplementary file 4. Top 10 markers from each granulosa subclusters.
- Supplementary file 5. Secreted markers expressed in granulosa cells varying with the estrous cycle.
- Transparent reporting form
- Source data 1. qPCR experiments individual p values.
- Source data 2. ELISA experiments individual p values.

### Data availability
Sequencing data have been deposited in Open Science Framework and the Broad Institute Single Cell Portal under study number SCP1914.

The following datasets were generated:

| Author(s) | Year | Dataset title | Dataset URL | Database and Identifier |
|---|---|---|---|---|
| David P | 2022 | Single Cell Sequencing of the Mouse Ovary in Diverse Reproductive States | https://osf.io/924fz/ | Open Science Framework, 924fz |
| David P | 2022 | A single cell atlas of the cycling murine ovary | https://singlecell.broadinstitute.org/single_cell/study/SCP1914/a-single-cell-atlas-of-the-cycling-murine-ovary | Broad Institute Single Cell Portal, SCP1914 |

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
