## [Editor Report]

This manuscript presents an important and useful dataset for understanding cellular and transcriptional dynamics during the estrous cycle in mice. Using single-cell RNA sequencing, the authors' data is compelling, providing new marker genes for different cell types. These data will be useful for understanding ovarian biology and will be of interest to biologists studying other tissues.

---

## [Decision Letter]

**Decision letter after peer review:**

Thank you for submitting your article "A single cell atlas of the cycling murine ovary" for consideration by *eLife*. Your article has been reviewed by 3 peer reviewers, and the evaluation has been overseen by a Reviewing Editor and Marianne Bronner as the Senior Editor. The reviewers have opted to remain anonymous.

Essential revisions:

The major concerns were regarding some data analysis and a request for further validation of the RNA seq. data.

1. In the Monocle in Figure 5C-D, why are the periovulatory GC that is the visualization look very much like mural GC in a completely different branch than mural GC? I have the impression that either the clusters need to be refined/validated further or the Monocle is not providing meaningful results regarding lineage trajectories. Why would the "periovulatory" GCs be at the end of the CL branch? In addition, the population of periovulatory Runx1+ cells seems rather aspecific. I think the authors could increase the robustness of the data/evidence regarding this cluster. Particularly because they go on comparing periovulatory and antral GCs. Could at least one differentially expressed marker from the volcano plot be validated?

2. The author used tSNE for visualization of the generated scRNA seq dataset, which, according to my knowledge, is outdated for scRNA seq data visualization as its reproducibility has become an issue. Which version of the Seurat package does the author use? And also the other software information should be implemented. Therefore, I suggest the author reanalyze their dataset using an updated Seurat pipeline, and also reanalyze all of their data using UMAP.

3. It does not appear that Figure 2 represents a real "subcluster" analysis. Instead, these cells are from the original clustering performed on all cells. To get a better resolution as to the different cell types, it would be best to perform a subcluster analysis where these cells are first extracted from the total data set and then re-clustered using more relevant principal components (for example, see Niu and Spradling, 2020).

4. For the pseudotime analysis presented in Figure 4 the authors excluded the mitotic and putative atretic granulosa cells. What is the justification for this?

5. The reviewers also have many other suggestions that the authors should likely consider to clarify the manuscript for the broad audience of *eLife* readers.

*Reviewer #1 (Recommendations for the authors):*

I think this is a fantastic dataset, but I think it could be better explored. The relation between different stages of estrous/lactating and the histology/validation could have been better explored and the cell types more explicitly named. I am not sure the Monocle analysis is clarifying things, this is tricky to use and I would consider using other comparable methods to determine the trajectories.

I have some points that I would like to see clarified:

– How was it determined that the postpartum 10-day females were lactating or not-lactating?

– Could the authors color the cells per oestrus phase, postlactating, randomly next to Figure 1B? It would be very useful to understand how the different cells in the initial clustering are represented in the different groups, particularly to determine the periovulatory follicles. In that sense, it would be very helpful to have corresponding histology of the ovaries in the respective cycle phase from the beginning. This aspect of the manuscript should be better explored. Also, the ovary from the lactating/nonlactating could be provided.

– Could the authors clarify what cluster corresponds to theca externa? In the text, they mentioned the expression of Mfap5, but in Figure 2 they only mention smooth muscle cells. How are the population for the smooth muscle cells and theca externa connected and or differentiated? This is confusing the text and Figures. In other words: where are the theca externa cells in Figure 2? Perhaps the authors could provide fields with higher magnification, eventually in the Suppl Figure? Could the "smooth muscle cells" simply be the theca externa?

– In Figure 2 is also not clear where are the theca interna cells, although the authors do mention those in the text. This is confusing. Please refer to the theca interna in the text and Figure. Could there be two populations of Theca interna (steroidogenic) and the ones closer to the basement membrane? I don't see any evidence to call this 'immature' theca (see Figure 4C).

– Regarding the atretic granulosa cells: the follicle provided still looks rather healthy. Is there a second marker you could use together with Ghr to confirm that particular cluster corresponds to atretic GC? Or can you colocalise Ghr with a marker of atresia?

– In addition, the periovulatory marker chosen does not seem very specific. I would suggest the authors choose a better marker. Otherwise, the identity of this cluster remains inconclusive. Moreover, what is the evidence that this would be periovulatory follicles? Is this the phase of the cycle? This could be better explained/evidenced, perhaps using histology of ovaries corresponding to the respective stages.

– The 3 clusters of granulosa cells (Figure 3) in the corpus luteum are intriguing. I wonder whether the authors have an explanation for that. What are the oestrous stages the cells were observed in?

– I am not sure the Monocle results (Figure 4A-B) are informative because the theca are not separated in theca interna and externa. Hence to me, it is unclear how the mesenchymal clusters, if they are well determined and reflect biology, relate to each other.

– In the Monocle in Figure 5C-D, I have similar issues – why are the periovulatory GC that is the visualization look very much like mural GC in a completely different branch than mural GC? I have the impression that either the clusters need to be refined/validated further or the Monocle is not providing meaningful results regarding lineage trajectories. Why would the "periovulatory" GCs be at the end of the CL^-^branch? In addition, the population of periovulatory Runx1+ cells seems rather aspecific. I think the authors could increase the robustness of the data/evidence regarding this cluster. Particularly because they go on comparing periovulatory and antral GCs. Could at least one differentially expressed marker from the volcano plot be validated?

– I am not sure the Monocle analysis is clarifying things, this is tricky to use and I would consider using other comparable methods to determine the trajectories.

– In Figure 4C: if the authors claim that the stroma is more medullary or cortex. In the Results, they do not visualise it. Hence, I would remove the claim from the Results but keep it in the Discussion.

– Results on the OSE should be shown in the Results section, now they are part of the Discussion? Where are those cells? What clusters? Etc. This population is very interesting/elusive and should be given proper attention/validation in the Results. In addition, the authors mention no differences between the estrous states: this should be clearly shown in the Results.

– The Discussion is too long, in particular, the discussion on the biomarkers reads more like a review than a discussion of your results. This should be shortened and made concise.

*Reviewer #3 (Recommendations for the authors):*

1. The introduction is very brief. It would be improved with a more thorough description of the estrus cycle beyond simply naming the stages. For example, how many days is the estrus cycle in the mouse? This would make the paper more accessible to non-experts.

2. Data availability- The authors have deposited the count matrix of the combined inDROP scRNA-seq dataset at Open Science Framework. However, for this dataset to serve as a useful resource for the wider community, the processed data should be uploaded to the Broad Institute Single Cell Portal (https://singlecell.broadinstitute.org).

3. The abstract indicates that in addition to ovaries isolated from the four stages of estrus, scRNA-seq was also performed on ovaries isolated from lactating or non-lactating 10 days postpartum mice, and from randomly cycling mice. However, cells from these later samples were never presented or discussed. It is therefore not clear what cells were used in the analysis. To make this clear, Figure 1 (or a supplemental figure) should include a tSNE plot where the cells are color-coded based on their sample origin (i.e. like Figure 5A).

4. Figure 1 – Given that this is the first figure, it would be good to show a tSNE plot that identifies which cells are from which library.

5. Figure 2 – It does not appear that Figure 2 represents a real "subcluster" analysis. Instead, these cells are from the original clustering performed on all cells. To get a better resolution as to the different cell types, it would be best to perform a subcluster analysis where these cells are first extracted from the total data set and then re-clustered using more relevant principal components (for example, see Niu and Spradling, 2020).

6. For the data presented in Figure 2, it is not entirely clear what is novel and what is already known. There are no references to the genes used to identify the cell sub-types types. References that are cited in the tables are not listed in the reference list. These references should, at a minimum, be listed in a supplemental text and it should be made clear in the text what genes are novel.

7. In addition to the markers chosen for specificity (e.g. Mafap5), it would be nice to see representative UMAPS (or dot plots) showing standard cell-type markers (e.g. Tcf21, Dcn, Notch3, Cxcl12). This could be a supplemental figure.

8. The description of the mesenchyme cell subpopulations is confusing as the authors do not use consistent terminology. A more concise definition of theca interna, theca externa, mature theca and immature theca as it relates to theca 1 and 2, stroma 1 and 2, smooth muscle would make the paper more accessible to the non-expert.

9. Line 196: Text refers to "dividing mesenchyme (8%) as seen in Figure 2A" but this population is not labeled. Which subcluster of cells is this referring to?

10. Line 204: Confusing to use theca externa when Figure 2C is labeled smooth muscle. Defining theca interna vs. theca externa, as mentioned above, would help.

11. Line 206: "…whereas Hhip was expressed in theca interna and immature theca (Figure S2A, B)." What is the difference between theca 1 and immature theca? Are these the same?

12. Figure 2 FigSup2: hard to see Acta2-Mfap5 overlap in Figure S2B. A higher magnification image would be helpful. It looks like there are two distinct cell layers, not the same cells. Also, Hhip expressing cells appear to be located outside of the Acta2-expressing cells, but Hhip is referred to as marking theca interna while Acta2 is said to mark theca externa. This needs to be explained.

13. Figure 2C – To make interpretation of the expression patterns more accessible to the non-expert, I suggest labeling some structures in these figure panels (e.g. oocyte, granulosa cells, cortical, medullary).

14. What cell type are the mitotic granulosa cells most similar to? Antral/mural or preantral/cumulus? Or are they an intermediate between these two? Subcluster analysis, as mentioned above, may help to better define their relationship.

15. Line 228: What is the justification for identifying these as atretic? What is the significance of Ghr to atresia? References would help.

16. Fan et al., used the following gene expression signatures to distinguish cumulus from mural GC's: cumulus GC (VCANhigh/FSThigh/IGFBP2high/HTRA1high/INHBBhigh/IHHhigh); mural GC (WT1low/EGR4low/KRT18high/CITED2high/LIHPhigh/AKIRIN1high). However, in Table S5 Fst and Inhbb are listed as markers for mural, not cumulus cells. This should be explained.

17. For the pseudotime analysis presented in Figure 4 the authors excluded the mitotic and putative atretic granulosa cells. What is the justification for this?

18. Figure 4D and E – Why are the preovulatory cells at the terminus of the pseudotime with CL2, while CL1 and CL3 cells positioned along the root. This does not match expectations. The discussion suggests that the developmental progression is PO>CL2>CL1>CL3, but this is not supported by the trajectory analysis. This calls into question the usefulness of the pseudotime analysis. It is mentioned in the discussion that instead of the different CL clusters representing a developmental progression, they are instead distinct cell types within a CL. This could be resolved with double RNA in situ hybridization using CL cluster-specific genes.

19. Figure 5A – It would be nice to show 4 separate plots for each of the stages. Hard to see this on a single plot. Perhaps 4 smaller panels next to A. It would also be helpful to label the different cell sub-types so the reader does not need to refer back to Figure 3.

20. Figure 5B – It would be helpful to label this panel "Proestrus-Estrus," following that in Figure S5.

21. Line 270 – reference to Figure S5D should be Figures 5E.

22. Figure S5D – This figure panel is not mentioned in the text.

23. Line 270: "…involved in wound repair during ovulation." This needs a reference.

24. It is stated that the motivation for identifying stage-specific secreted biomarkers was to identify markers that would be useful for staging in assisted reproduction and other applications in reproductive medicine. This begs the question if Prss35, Nppc and Tinagl1 are also expressed in human ovaries? Was this part of the reason they were selected?

25. Figure 6B: I think panel B should be first since it is from the scRNA-seq data set and then A is validation. I would also add Lhcgr and Pgr to the dot plot in B. Also, it makes more sense to put Inhba next to Nppc.

26. Figure 6A: The graphs in A are somewhat randomly organized, which makes it unnecessarily complicated. These should be reorganized to group factors high in P vs. E together and those high in E vs. P together.

27. Figure 6C: Follicle stages should be labeled. Mural vs. Cumulus cell should be labeled. Would be best to show both follicles that are labeled and ones that are not labeled to emphasize that this is a stage-specific expression. I cannot see Nppc expression. Might be helpful to have a higher magnification or arrows pointing to expressing cells.

28. Line 234: Run-on sentence: "Early pre-antral follicles…"

29. Line 384- Fan et al., reference should be deleted here.

---

## [Author Response]

Essential revisions:The major concerns were regarding some data analysis and a request for further validation of the RNA seq. data.1. In the Monocle in Figure 5C-D, why are the periovulatory GC that is the visualization look very much like mural GC in a completely different branch than mural GC? I have the impression that either the clusters need to be refined/validated further or the Monocle is not providing meaningful results regarding lineage trajectories. Why would the "periovulatory" GCs be at the end of the CL branch?

We agree that the pseudo-time analysis we presented using monocle appeared counter intuitive. Furthermore, other pseudotime analyses gave similar results. This is because the lineage from preantral granulosa cell to cumulus granulosa cells of antral follicles is linear. In contrast, mural granulosa cells of antral follicles have a distinct trajectory representing a branching point leading to luteal differentiation. However to luteinizing cell state is unique, and more similar to luteal cells of the corpus luteum than to antral mural granulosa. This creates a discontinuous trajectory of granulosa lineages in pseudotime, where the transcriptional response to the LH surge appears as a terminal state, when it is in fact transient. Therefore, in an effort to improve clarity, we have removed the pseudotime analysis in favor of more concise explanation of these lineages and states in the discussion.

In addition, the population of periovulatory Runx1+ cells seems rather aspecific. I think the authors could increase the robustness of the data/evidence regarding this cluster. Particularly because they go on comparing periovulatory and antral GCs.

To improve the robustness of these conclusions we have replaced Runx1 by Oxtr which is more specific to this cell type (See Figure 3D and 3-supplement 1D). In addition, we refer to a swath of previously validated markers which are very specific to Graafian follicles.

Could at least one differentially expressed marker from the volcano plot be validated?

We validated several of the markers shown in the volcano plot as presented in in Fig5E.

2. The author used tSNE for visualization of the generated scRNA seq dataset, which, according to my knowledge, is outdated for scRNA seq data visualization as its reproducibility has become an issue. Which version of the Seurat package does the author use? And also the other software information should be implemented. Therefore, I suggest the author reanalyze their dataset using an updated Seurat pipeline, and also reanalyze all of their data using UMAP.

Following the reviewers’ advice, we reanalyzed our dataset and presented the dimensional reductions using UMAP. Information regarding the Seurat package and R version have been added to the material and methods section.

3. It does not appear that Figure 2 represents a real "subcluster" analysis. Instead, these cells are from the original clustering performed on all cells. To get a better resolution as to the different cell types, it would be best to perform a subcluster analysis where these cells are first extracted from the total data set and then re-clustered using more relevant principal components (for example, see Niu and Spradling, 2020).

When reanalyzing all our data with UMAP we re-clustered each cell type independently, as Niu and Spradling.

4. For the pseudotime analysis presented in Figure 4 the authors excluded the mitotic and putative atretic granulosa cells. What is the justification for this?

Given the reviewers’ comments we have elected to remove the pseudotime trajectory analysis in favor of more concise interpretation of lineages.

5. The reviewers also have many other suggestions that the authors should likely consider to clarify the manuscript for the broad audience of eLife readers.

We thank the reviewers for their constructive comments regarding the manuscript. We have edited the presentation of our findings to make it more accessible to the broad readership of *eLife* and anticipate that this manuscript will provide a useful resource for the research community regarding the dynamic changes in transcriptome at the single cell level in the ovary as a function of the estrous cycle.

Reviewer #1 (Recommendations for the authors):I think this is a fantastic dataset, but I think it could be better explored. The relation between different stages of estrous/lactating and the histology/validation could have been better explored and the cell types more explicitly named. I am not sure the Monocle analysis is clarifying things, this is tricky to use and I would consider using other comparable methods to determine the trajectories.I have some points that I would like to see clarified:– How was it determined that the postpartum 10-day females were lactating or not-lactating?

The pups were removed from the female on the day they were born, hence when the ovaries were harvested at post-partum day 10 while the mice were not-lactating. The lactating mice remained with their pups at post-partum day 10. The text has been modified to clarify the method (lines 77-80).

– Could the authors color the cells per oestrus phase, postlactating, randomly next to Figure 1B? It would be very useful to understand how the different cells in the initial clustering are represented in the different groups, particularly to determine the periovulatory follicles. In that sense, it would be very helpful to have corresponding histology of the ovaries in the respective cycle phase from the beginning. This aspect of the manuscript should be better explored. Also, the ovary from the lactating/nonlactating could be provided.

We added a dimensional reduction showing the distribution of the cells depending on the stage of the estrous cycle in which they have been collected (Figure 1B). The corresponding representative histology for each phase of the estrous cycle as well as day 10 post-partum lactating and non-lactating is illustrated on Figure S1B.

– Could the authors clarify what cluster corresponds to theca externa? In the text, they mentioned the expression of Mfap5, but in Figure 2 they only mention smooth muscle cells. How are the population for the smooth muscle cells and theca externa connected and or differentiated? This is confusing the text and Figures. In other words: where are the theca externa cells in Figure 2? Perhaps the authors could provide fields with higher magnification, eventually in the Suppl Figure? Could the "smooth muscle cells" simply be the theca externa?

We added some clarification of the cell types present in the theca externa which includes the Smooth muscle cells which we validated in greater detail (lines 220-229). We also provided in supplemental figure 2 – supplement 1 a higher magnification of Hhip-Acta2 and well as Mfap5-Acta2 to be able to differentiate smooth muscle cells of the theca externa from the early theca cells of the theca interna.

– In Figure 2 is also not clear where are the theca interna cells, although the authors do mention those in the text. This is confusing. Please refer to the theca interna in the text and Figure. Could there be two populations of Theca interna (steroidogenic) and the ones closer to the basement membrane? I don't see any evidence to call this 'immature' theca (see Figure 4C).

The theca cell clusters were renamed Steroidogenic and early theca. We provided a new micrograph of Hhip in situ hybridization to show expression in “early” theca cells surrounding preantral follicles, while Cyp17a1 is expressed in “steroidogenic” theca cells surrounding antral follicles in Figures 2C and Figure 2 – supplement 1A, B.

– Regarding the atretic granulosa cells: the follicle provided still looks rather healthy. Is there a second marker you could use together with Ghr to confirm that particular cluster corresponds to atretic GC? Or can you colocalise Ghr with a marker of atresia?

We provided a better representative example of atretic follicle. To confirm that this cluster corresponds to atretic granulosa cells, we provided in Figure S3A a feature plots of other markers known to be expressed in apoptotic cells that colocalized with Ghr.

– In addition, the periovulatory marker chosen does not seem very specific. I would suggest the authors choose a better marker. Otherwise, the identity of this cluster remains inconclusive. Moreover, what is the evidence that this would be periovulatory follicles? Is this the phase of the cycle? This could be better explained/evidenced, perhaps using histology of ovaries corresponding to the respective stages.

We replaced Runx1 by Oxtr as the representative periovulatory marker, which appears more specific by in-situ hybridization (Figures3C and S3D). The classification of this cell type is further supported by a fullyreferenced list of markers in Table S5. Furthermore, this cluster is only found in mice as estrous as indicated in Figure 5A/B, which is consistent spatially and temporally with mural granulosa cells of periovulatory graafian follicles responding to LH and initiating luteinization.

– The 3 clusters of granulosa cells (Figure 3) in the corpus luteum are intriguing. I wonder whether the authors have an explanation for that. What are the oestrous stages the cells were observed in?

Following reanalysis of our dataset with the latest version of Seurat and the dimensional representation method to UMAP we now have two corpus luteum (CL) clusters instead of 3. We hypothesize that these two clusters correspond to active and regressing CL respectively. We confirmed this hypothesis by evaluating the level of expression of a proliferative marker such as Top2a and a cell cycle arrest such as Cdkn1a (Figure S3B). We further validated the spatial colocalization of these markers, with large healthy corpora lutea expressing Top2a while smaller luteolysing CLs express the cell cycle inhibitor and senescence marker Cdkn1a along with a cohort of validated markers shown in FigS3E and Table S4.

– I am not sure the Monocle results (Figure 4A-B) are informative because the theca are not separated in theca interna and externa. Hence to me, it is unclear how the mesenchymal clusters, if they are well determined and reflect biology, relate to each other.

We have removed the pseudotime analysis from the manuscript.

– In the Monocle in Figure 5C-D, I have similar issues – why are the periovulatory GC that is the visualization look very much like mural GC in a completely different branch than mural GC? I have the impression that either the clusters need to be refined/validated further or the Monocle is not providing meaningful results regarding lineage trajectories.

We agree with the reviewer. This figure was removed.

Why would the "periovulatory" GCs be at the end of the CL^-^branch? In addition, the population of periovulatory Runx1+ cells seems rather aspecific. I think the authors could increase the robustness of the data/evidence regarding this cluster. Particularly because they go on comparing periovulatory and antral GCs. Could at least one differentially expressed marker from the volcano plot be validated?

As discussed above, the position of the periovulatory granulosa cells in the monocle visual representation of pseudotime is a function of the LH response signature state, rather than a linear differentiation trajectory, which complicates the interpretation of those plots, as the cells then revert along the same axis to a luteinized state with reduced acute response to LH. To clarify the manuscript, we have removed the pseudotime plot. The classification of periovulatory GC is further supported by a fully-referenced list of markers in Table S5.

– I am not sure the Monocle analysis is clarifying things, this is tricky to use and I would consider using other comparable methods to determine the trajectories.

We agree with the reviewer. This figure was removed.

– In Figure 4C: if the authors claim that the stroma is more medullary or cortex. In the Results, they do not visualise it. Hence, I would remove the claim from the Results but keep it in the Discussion.

We agree with the reviewer. Following reanalyzing our data with UMAP we couldn’t identify unique dichotomous markers to differentiate each cluster. However, we found Ennp2 to be differentially expressed: present in one cluster (fibroblast-like stroma) and completely absent from the other (steroidogenic stroma). These results are now part of figure 2 and figure 2 – supplement 1.

– Results on the OSE should be shown in the Results section, now they are part of the Discussion? Where are those cells? What clusters? Etc. This population is very interesting/elusive and should be given proper attention/validation in the Results. In addition, the authors mention no differences between the estrous states: this should be clearly shown in the Results.

The epithelium cluster is composed of the ovarian surface epithelium cells. We now explore this cluster deeper in Figure 4. We showed that this cluster could be subdivided in two clusters one of them being characterized by a very strong expression of proliferation markers and being composed almost exclusively of granulosa cells collected during the estrous phase of the cycle, consistent with a transient amplification of these cells coinciding with ovulatory wound repair.

– The Discussion is too long, in particular, the discussion on the biomarkers reads more like a review than a discussion of your results. This should be shortened and made concise.

We thank the reviewer for this suggestion, we significantly revised the discussion to clarify our results in the context of ovarian biology and to make it more concise.

Reviewer #3 (Recommendations for the authors):1. The introduction is very brief. It would be improved with a more thorough description of the estrus cycle beyond simply naming the stages. For example, how many days is the estrus cycle in the mouse? This would make the paper more accessible to non-experts.

We thank the reviewer for this suggestion, we significantly rewrote the introduction and discussion to improve clarity and make the manuscript more accessible to the broad readership of *eLife*. We hope this dataset will become a valuable resource to the community to investigate dynamic cell states associated with estrous cycling in the ovary.

2. Data availability- The authors have deposited the count matrix of the combined inDROP scRNA-seq dataset at Open Science Framework. However, for this dataset to serve as a useful resource for the wider community, the processed data should be uploaded to the Broad Institute Single Cell Portal (https://singlecell.broadinstitute.org).

We have deposited the data set and source code on the Broad Institute Single Cell Portal under study number SCP1914.

3. The abstract indicates that in addition to ovaries isolated from the four stages of estrus, scRNA-seq was also performed on ovaries isolated from lactating or non-lactating 10 days postpartum mice, and from randomly cycling mice. However, cells from these later samples were never presented or discussed. It is therefore not clear what cells were used in the analysis. To make this clear, Figure 1 (or a supplemental figure) should include a tSNE plot where the cells are color-coded based on their sample origin (i.e. like Figure 5A).

We added panels to Figure 1B and 5A/B to represent the distribution of cells depending on the stage of the estrous cycle. We also illustrated the morphology of the ovary depending on the stage of the estrous cycle and lactation state on Figure S1B. Finally, we analyzed the post-partum lactation data in Figure S3E and discuss it in the results (lines 272-275).

4. Figure 1 – Given that this is the first figure, it would be good to show a tSNE plot that identifies which cells are from which library.

We now reran the analysis to show UMAP dimensional reduction instead of tSNE and updated all figures of the manuscript. We also added a panel in Figure 1B to identify which cell are coming from which library.

5. Figure 2 – It does not appear that Figure 2 represents a real "subcluster" analysis. Instead, these cells are from the original clustering performed on all cells. To get a better resolution as to the different cell types, it would be best to perform a subcluster analysis where these cells are first extracted from the total data set and then re-clustered using more relevant principal components (for example, see Niu and Spradling, 2020).

We thank the reviewer for this suggestion. We re-clustered each object and displayed the results using UMAP projections and revised all the figures of the manuscript accordingly.

6. For the data presented in Figure 2, it is not entirely clear what is novel and what is already known. There are no references to the genes used to identify the cell sub-types types. References that are cited in the tables are not listed in the reference list. These references should, at a minimum, be listed in a supplemental text and it should be made clear in the text what genes are novel.

We thank the reviewer for pointing this out. The top markers for each cluster/cell type are presented in Tables S2, S3, and S4. The markers that have previously been described are fully referenced in these tables, and those citations are now part of the manuscript’s bibliography.

7. In addition to the markers chosen for specificity (e.g. Mafap5), it would be nice to see representative UMAPS (or dot plots) showing standard cell-type markers (e.g. Tcf21, Dcn, Notch3, Cxcl12). This could be a supplemental figure.

We thank the reviewer for this suggestion. A DotPlot was used to show the expression of the suggested and validated cell-type markers, which is now located in Figure S2D.

8. The description of the mesenchyme cell subpopulations is confusing as the authors do not use consistent terminology. A more concise definition of theca interna, theca externa, mature theca and immature theca as it relates to theca 1 and 2, stroma 1 and 2, smooth muscle would make the paper more accessible to the non-expert.

The new clustering analysis and nomenclature clarify the mesenchymal cell types. The description of the theca interna and externa, and their cellular composition is now discussed more clearly in the results (lines 230-244) and discussion (lines 297- 417), and visualized in Figure 1 – supplement 1. We also added a zoomed picture of the theca stained with SMA and Mfap5 or Hhip to help the reader differentiate smooth muscles of the theca externa to the early theca cells of the theca interna depending on expression and position in figure 2 – supplement 1B.

9. Line 196: Text refers to "dividing mesenchyme (8%) as seen in Figure 2A" but this population is not labeled. Which subcluster of cells is this referring to?

This population no longer represent a distinct cluster in our new analysis.

10. Line 204: Confusing to use theca externa when Figure 2C is labeled smooth muscle. Defining theca interna vs. theca externa, as mentioned above, would help.

The description of the theca interna and externa, and their cellular composition is now discussed more clearly in the results (lines 230-244) and discussion (lines 397-417), and visualized in Figure 1supplement 1A

11. Line 206: "…whereas Hhip was expressed in theca interna and immature theca (Figure S2A, B)." What is the difference between theca 1 and immature theca? Are these the same?

We adopted a consistent nomenclature reflecting the updated subclustering analysis. The theca clusters have been renamed early theca, steroidogenic theca, and smooth muscle (of the theca externa) and the text was modified accordingly.

12. Figure 2 FigSup2: hard to see Acta2-Mfap5 overlap in Figure S2B. A higher magnification image would be helpful. It looks like there are two distinct cell layers, not the same cells. Also, Hhip expressing cells appear to be located outside of the Acta2-expressing cells, but Hhip is referred to as marking theca interna while Acta2 is said to mark theca externa. This needs to be explained.

We added higher magnification of Hhip and Mfap5 staining colocalization with Acta2 in Figure 2 – supplement 1B. While Hhip is expressed in the early theca, of the theca interna, Mfap5 is expressed in smooth muscle cells of the theca externa and colocalizes with Acta2, a general marker of smooth muscle.

13. Figure 2C – To make interpretation of the expression patterns more accessible to the non-expert, I suggest labeling some structures in these figure panels (e.g. oocyte, granulosa cells, cortical, medullary).

We thank the reviewer for this suggestion. The general histology of ovarian structures across diverse reproductive states are now shown in supplemental figure 1 – supplement 1A.

14. What cell type are the mitotic granulosa cells most similar to? Antral/mural or preantral/cumulus? Or are they an intermediate between these two? Subcluster analysis, as mentioned above, may help to better define their relationship.

The mitotic granulosa cells cluster was composed of cells that were closely related to both the antral mural and the preantral-cumulus granulosa (see Author response image 2 the expression of both markers Mro (red) and Kctd14(green) in the mitotic cluster). While these clusters may have been differentiated with a higher cluster resolution, we chose to keep the resolution consistent across all subclustering.

**Author response image 1. sa2fig1:** 

15. Line 228: What is the justification for identifying these as atretic? What is the significance of Ghr to atresia? References would help.

We propose Ghr as a new marker for atretic granulosa cells. We confirmed the identification of atretic cluster based on expression of apoptotic markers such as Pik3ip1, Nupr1 and Gadd45a presented on Figure S3A along with additional previously validated markers which are referenced in Tables S4. Moreover, we provided a better representative example of an atretic follicle with Ghr in-situ hybridization in Figure 3C.

16. Fan et al., used the following gene expression signatures to distinguish cumulus from mural GC's: cumulus GC (VCANhigh/FSThigh/IGFBP2high/HTRA1high/INHBBhigh/IHHhigh); mural GC (WT1low/EGR4low/KRT18high/CITED2high/LIHPhigh/AKIRIN1high). However, in Table S5 Fst and Inhbb are listed as markers for mural, not cumulus cells. This should be explained.

While we investigated all markers discussed in the Fan et al., publication, many were not conserved from human to mouse (see Author response image 2). We elaborate on some of these inter-species’ discrepancies in the discussion (lines 414-417).

17. For the pseudotime analysis presented in Figure 4 the authors excluded the mitotic and putative atretic granulosa cells. What is the justification for this?

Given that the monocle pseudotime trajectories of granulosa cells were difficult to interpret, we have removed this analysis from the manuscript.

18. Figure 4D and E – Why are the preovulatory cells at the terminus of the pseudotime with CL2, while CL1 and CL3 cells positioned along the root. This does not match expectations. The discussion suggests that the developmental progression is PO>CL2>CL1>CL3, but this is not supported by the trajectory analysis. This calls into question the usefulness of the pseudotime analysis. It is mentioned in the discussion that instead of the different CL clusters representing a developmental progression, they are instead distinct cell types within a CL. This could be resolved with double RNA in situ hybridization using CL cluster-specific genes.

We agree that the pseudo-time analysis we presented using monocle appears counter intuitive. This is because the pseudotime progression from mural granulosa cells of antral follicles to periovulatory follicle granulosa cells, to active CL to regressing CL does not appear linear. Therefore, in an effort to improve clarity, we have removed the pseudotime analysis in favor of more concise explanation of these lineages and states in the discussion (line 378-382).

19. Figure 5A – It would be nice to show 4 separate plots for each of the stages. Hard to see this on a single plot. Perhaps 4 smaller panels next to A. It would also be helpful to label the different cell sub-types so the reader does not need to refer back to Figure 3.

We thank the reviewer for this suggestion. We added a new figure (Figure 5B) showing 4 separate plots for each of the stages.

20. Figure 5B – It would be helpful to label this panel "Proestrus-Estrus," following that in Figure S5.

We thank the reviewer for bringing this to our attention, we have added a clear label.

21. Line 270 – reference to Figure S5D should be Figures 5E.

We thank the reviewer for bringing this to our attention, we fixed this mistake.

22. Figure S5D – This figure panel is not mentioned in the text.

We have added this to the text (line 317).

23. Line 270: "…involved in wound repair during ovulation." This needs a reference.

We have added references to the text (lines 317).

24. It is stated that the motivation for identifying stage-specific secreted biomarkers was to identify markers that would be useful for staging in assisted reproduction and other applications in reproductive medicine. This begs the question if Prss35, Nppc and Tinagl1 are also expressed in human ovaries? Was this part of the reason they were selected?

To prioritize these markers, we identified factors specifically secreted by ovarian granulosa in our dataset, and predominantly expressed by the gonads in the human GTEX database. We have added selection details to the text (lines 320-324).

25. Figure 6B: I think panel B should be first since it is from the scRNA-seq data set and then A is validation. I would also add Lhcgr and Pgr to the dot plot in B. Also, it makes more sense to put Inhba next to Nppc.

We thank the reviewer for the suggestion. We modified Figure 6B accordingly.

26. Figure 6A: The graphs in A are somewhat randomly organized, which makes it unnecessarily complicated. These should be reorganized to group factors high in P vs. E together and those high in E vs. P together.

We thank the reviewer for the suggestion. We modified Figure 6A accordingly.

27. Figure 6C: Follicle stages should be labeled. Mural vs. Cumulus cell should be labeled. Would be best to show both follicles that are labeled and ones that are not labeled to emphasize that this is a stage-specific expression. I cannot see Nppc expression. Might be helpful to have a higher magnification or arrows pointing to expressing cells.

We thank the reviewer for the suggestion. We labelled the structures in Fig6C and replaced the Nppc panel with a picture with a clearer stain.

28. Line 234: Run-on sentence: "Early pre-antral follicles…"

We thank the reviewer for bringing this to our attention, we fixed this mistake.

29. Line 384- Fan et al., reference should be deleted here.

We thank the reviewer for bringing this to our attention, we fixed this mistake.